# Position: Constants are Critical in Regret Bounds for Reinforcement Learning

**Simone Drago** [1]   **Marco Mussi** [1]   **Alberto Maria Metelli** [1]

## Abstract

Mainstream research in theoretical RL is currently focused on designing online learning algorithms with regret bounds that match the corresponding regret lower bound up to multiplicative constants (and, sometimes, logarithmic terms). In this position paper, we constructively question this trend, arguing that algorithms should be designed to at least minimize the amount of unnecessary exploration, and we highlight the significant role constants play in algorithms' actual performances. This trend also exacerbates the misalignment between theoretical researchers and practitioners. As an emblematic example, we consider the case of regret minimization in finite-horizon tabular MDPs. Starting from the well-known `UCBVI` algorithm, we improve the bonus terms and the corresponding regret analysis. Additionally, we compare our version of `UCBVI` with both its original version and the state-of-the-art `MVP` algorithm. Our empirical validation successfully demonstrates how improving the multiplicative constants has significant positive effects on the actual empirical performances of the algorithm under analysis. This raises the question of whether ignoring constants when assessing whether algorithms match is the proper approach.

## 1. Introduction

Reinforcement Learning (RL, Sutton & Barto, 2018) has emerged as a powerful methodology for addressing sequential decision-making problems under uncertainty. Besides the successful applications of RL in the last decades, including robotic locomotion (Kober et al., 2013), continuous system control (Schulman et al., 2015; Lillicrap et al., 2016; Haarnoja et al., 2018), autonomous driving (Kiran et al., 2021), and games (Mnih et al., 2015; Silver et al.,

2017), RL has been subject to a lot of attention from the research community from the theoretical standpoint, obtaining significant understanding on its fundamental statistical challenges (Azar et al., 2017; Domingues et al., 2021).

Taking inspiration from the *multi-armed bandit* (MAB) literature (Lattimore & Szepesvári, 2020), theoretical research in RL has focused on characterizing *online* RL algorithms through the *cumulative regret*, which represents the suboptimality of the played policy summed over a given horizon of interaction. This performance index is particularly meaningful as it quantifies *how quickly* the learning algorithm converges to the optimal policy. The conventional path in this literature consists of $(i)$ characterizing the statistical complexity of the problem by deriving *regret lower bounds* and $(ii)$ designing and analyzing algorithms to provide *regret upper bounds*. The comparison between upper bounds and lower bounds allows to establish whether the problem is *closed*. It is conventionally accepted that a problem is closed when upper and lower bounds match in the characteristic parameters of the problem, apart from constant (and, sometimes, logarithmic) multiplicative factors.

Consider, for instance, the emblematic case of *finite-horizon tabular RL*, where we have a well-established lower bound in the order of $\Omega(\sqrt{HSAT})$, where $S$ is the number of states, $A$ is the number of actions, $H$ is the horizon of the episode, and $T = HK$ where $K$ is the number of episodes (Domingues et al., 2021).[1] At present, we have learning algorithms that match this lower bound. The first approach that succeeded in achieving order-optimal regret is `UCBVI` (Azar et al., 2017), combining the classical value iteration approach with the famous *optimism in the face of uncertainty* mechanism, borrowed from bandits. The algorithm is very intuitive (although its analysis is quite convoluted) and manages to achieve the same order of $\widetilde{\mathcal{O}}(\sqrt{HSAT})$ under the assumption that the time horizon $T$ is sufficiently large ($T \geqslant \mathcal{O}(H^3S^3A)$).[2] This latter limitation has been recently overcome by `MVP` (Zhang et al.,

[1]Politecnico di Milano, Milan, Italy.
Correspondence to: Simone Drago <simone.drago@polimi.it>.

*Proceedings of the $42^{nd}$ International Conference on Machine Learning*, Vancouver, Canada. PMLR 267, 2025. Copyright 2025 by the author(s).

---

[1]In this paper, we consider stage-independent transition probabilities. For stage-dependent ones, the lower bound presents an additional $\sqrt{H}$ multiplicative term. From the algorithmic perspective, we can easily adapt the upper bound results by considering an MDP with an augmented state space of size $SH$.

[2]The $\mathcal{O}(\cdot)$ notation ignores constants, while $\widetilde{\mathcal{O}}(\cdot)$ also ignores logarithmic factors.

2024) at the price of a less intuitive algorithm that is forced to discard previously collected sample because of the use of a *doubling trick*. Thus, in the sense defined above, we can say that *finite-horizon tabular RL is a closed problem*.

In this paper, we constructively question this statement. Indeed, as intuition suggests, **ignoring multiplicative constants when evaluating whether an algorithm matches the problem's lower bound may lead to disappointing results when *using* such an algorithm**. Constants matter in two key moments:

(*i*) *Algorithm design*: the exploration strategy of the algorithm should be designed to enforce the minimal exploration required to achieve the desired regret performance. For example, in an optimistic algorithm like `UCBVI` (Azar et al., 2017), the exploration bonuses should be as small as possible;

(*ii*) *Algorithm analysis*: the analysis of the algorithm should be conducted using the most accurate analytical tools possible in order to obtain a *regret upper bound* that closely approximates the *actual regret* suffered by the algorithm.

While (*ii*) impacts the regret bounds and, to a certain extent, can be considered a secondary requirement, (*i*) has far more dramatic effects. Indeed, an inaccurate and overly conservative design of the exploration strategy significantly affects the algorithm's empirical performances, wasting samples and ultimately leading to unnecessary over-exploration.

To support these claims, in this paper, we will consider as a case study the `UCBVI` algorithm (Azar et al., 2017), and we compare it with the state of the art for this setting, `MVP` (Zhang et al., 2024). `MVP` employs a doubling trick, that, while convenient in terms of regret analysis, negatively affects the efficiency of the algorithm in practice, leading to an inevitable over-exploration and a higher regret. We will provide an improvement of `UCBVI`, in its advanced solution with Bernstein-Freedman bonus, with the goal of (*i*) deriving an exploration bonus that is as tight as possible and (*ii*) conducting a regret upper bound analysis that generates the smallest constants possible. We show empirically how a more careful design of the exploration bonus delivers dramatic improvements in the empirical performance with a reduction of the regret by a factor of $1.87$ on average.

Additionally, in Appendix A, we support our discussion by reporting works from the MAB literature that devise algorithms optimizing both constants and lower order terms, thus demonstrating an interest in their empirical performance which is well reflected in their experimental validation.

## 2. Preliminaries

In this section, we briefly introduce the notation and concepts employed in the rest of this work.

**Notation.** Given a measurable set $\mathcal{X}$, we denote with $\Delta(\mathcal{X})$ the set of probability measures over $\mathcal{X}$. For $n \in \mathbb{N}$, we denote the set $\{1, \ldots, n\}$ as $[\![n]\!]$. We denote the L1 norm of a vector as $\|\cdot\|_1$.

**Markov Decision Processes.** An undiscounted, finite-horizon Markov Decision Process (MDP, Puterman, 1994) is a tuple $\mathcal{M} := (\mathcal{S}, \mathcal{A}, P, R, H)$, where $\mathcal{S}$ is the state space, $\mathcal{A}$ is the action space, $P : \mathcal{S} \times \mathcal{A} \to \Delta(\mathcal{S})$ represents the state transition probability, $R : \mathcal{S} \times \mathcal{A} \to \mathbb{R}$ represents the reward function, and $H \in \mathbb{N}_{>0}$ is the length of each episode.[3]

We assume the state space and the action space to be finite sets, and we denote their cardinalities as $|\mathcal{S}| = S$ and $|\mathcal{A}| = A$. We assume that the state transition probability and the reward do not depend on the stage. Moreover, we assume the reward to be deterministic, known, and bounded in $[0, 1]$.[4]

**Interaction with the Environment.** The agent interacts with the environment in a sequence of $K$ episodes. Denote as $x_{k,h}$ the state occupied by the agent at stage $h \in [\![H]\!]$ of episode $k \in [\![K]\!]$, and as $a_{k,h}^{\pi_k}$ the action played by the agent at stage $h$ of episode $k$ according to the policy $\pi_k$. We assume the policies to be deterministic and stage-dependent, i.e., $\pi : \mathcal{S} \times [\![H]\!] \to \mathcal{A}$.

The interaction of the $k$-th episode starts from state $x_{k,1} \in \mathcal{S}$, then, the agent selects which action to play as $a_{k,h}^{\pi_k} = \pi_k(x_{k,h}, h)$ for $h \in [\![H]\!]$, and observes a sequence of next-states and reward, until the end of the episode.

The function $V_h^\pi : \mathcal{S} \to \mathbb{R}$ denotes the value function at stage $h \in [\![H]\!]$, such that $V_h^\pi(x)$ represents the expected sum of the $H - h$ returns received under policy $\pi$ starting from state $x \in \mathcal{S}$. Under the assumptions above, there exists a deterministic policy $\pi^*$ which attains the best possible value function $V_h^*(x) := \sup_\pi V_h^\pi(x)$ for every state $x \in \mathcal{S}$. We measure the performance of a learning algorithm $\mathfrak{A}$ after $K$ episodes by means of the *cumulative regret*:

$$\text{Reg}(\mathfrak{A}, K) := \sum_{i=1}^{K} V_1^*(x_{i,1}) - V_1^{\pi_i}(x_{i,1}).$$

We denote as $T = KH$ the total number of interactions.

## 3. A Refined `UCBVI` Algorithm and Analysis

In this section, we consider the `UCBVI` algorithm, introduced in (Azar et al., 2017), for which we provide a more

---

[3] Let $x, y \in \mathcal{S}$ and $a \in \mathcal{A}$, we denote as $P(y|x, a)$ the probability of observing $y$ as the next state after playing action $a$ in state $x$, and $R(x, a)$ the reward obtained after playing action $a$ in state $x$.

[4] The assumption on the knowledge of the reward can be removed without relevant drawbacks on the algorithm's theoretical guarantees, as learning the stochastic transition model is more challenging than learning the reward function.

---

**Algorithm 1:** `UCBVI`.

1 **Initialize**: $N_k(x, a, y) = 0$, $N_k(x, a) = 0$, $N'_{k,h}(x) = 0$, $\forall(x, a, y) \in \mathcal{S} \times \mathcal{A} \times \mathcal{S}$

2          $Q_{0,h}(x, a) = H - h + 1$, $\forall(x, a, h) \in \mathcal{S} \times \mathcal{A} \times [\![H]\!]$

3 **for** $k \in [\![K]\!]$ **do**

4      Estimate $\hat{P}_k(y|x, a) = N_k(x, a, y)/N_k(x, a)$

5      Initialize $V_{k,H+1}(x) = 0$, $\forall x \in \mathcal{S}$

6      **for** $h = \{H, H - 1, \dots, 1\}$ **do**

7          **for** $x \in \mathcal{S}$ **do**

8              $Q_{k,h}(x, a) = \min\{Q_{k-1,h}(x, a), R(x, a) + \sum_{y \in \mathcal{S}} \hat{P}_k(y|x, a)V_{k,h+1}(y) + b_{k,h}(x, a)\}$, $\forall a \in \mathcal{A}$

9              $V_{k,h}(x) = \max_{a \in \mathcal{A}} Q_{k,h}(x, a)$

10          **end**

11      **end**

12      Agent observes state $x_{k,1}$

13      **for** $h \in [\![H]\!]$ **do**

14          Agent plays action $a_{k,h} \in \arg\max_{a \in \mathcal{A}} Q_{k,h}(x_{k,h}, a)$

15          Environment returns reward $r_{k,h}$ and next state $x_{k,h+1}$

16          Increment counters $N_k(x_{k,h}, a_{k,h}, x_{k,h+1})$, $N_k(x_{k,h}, a_{k,h})$, $N'_{k,h}(x_{k,h})$

17      **end**

18 **end**

---

compact (but equivalent) pseudocode in Algorithm 1. In Theorems 3.1 and 3.2, we provide results on the regret upper bound under the usage of bonuses defined via the *Chernoff-Hoeffding* and the *Bernstein-Freedman* inequalities, respectively. Our contribution consists of refining the analysis to obtain tighter bonuses and, as a direct consequence, tighter regret bounds.

**Regret Bounds.** We now state the results of the refined regret upper bound analysis. Let us start with the *Chernoff-Hoeffding* version, taking the opportunity to fix some typos of the original analysis of (Azar et al., 2017).

**Theorem 3.1** (Regret for `UCBVI` with Chernoff-Hoeffding bound). *Let $\delta \in (0, 1)$. Considering:*[5]

$$b_{k,h}(x, a) = \frac{2HL}{\sqrt{\max\{N_k(x, a), 1\}}},$$

*then, w.p. at least $1 - \delta$, the regret of `UCBVI-CH` is bounded by:*

$$\mathrm{Reg}(\mathtt{UCBVI\text{-}CH}, K) \leqslant 10eHL\sqrt{SAT} + \frac{8}{3}eH^2S^2AL^2,$$

*where $L = \ln(5HSAT/\delta)$. For $T \geqslant H^2S^3A$, this bound translates to $\tilde{\mathcal{O}}(H\sqrt{SAT})$.*

Theorem 3.1 should be compared to Theorem 1 of (Azar et al., 2017). Since the analysis is a refinement of the original analysis in terms of constants, the order of the regret does

not change between the two theorems. However, our analysis provides a smaller value for the constants.[6] Moreover, observe how the minimum value of $T$ for which the regret bound holds according to our analysis is $H$ times higher than the one reported in the original theorem. This is due to the fact that the minimum $T$ in the statement of Theorem 1 of (Azar et al., 2017) is incorrect, although the derivation in the appendix provides the same minimum value of $T$ we obtain.

Let us move to the *Bernstein-Freedman* bonus.

**Theorem 3.2** (Regret for `UCBVI` with Bernstein-Freedman bound). *Let $\delta \in (0, 1)$. Considering:*[5]

$$b_{k,h}(x, a) = \underbrace{\sqrt{\frac{4L\, \mathbb{V}\mathrm{ar}_{y \sim \hat{P}_k(\cdot|x,a)}(V_{k,h+1}(y))}{\max\{N_k(x, a), 1\}}}}_{(A)} +$$

$$+ \underbrace{\frac{7HL}{3\max\{N_k(x, a) - 1, 1\}}}_{(B)} +$$

$$+ \underbrace{\sqrt{\frac{4\sum_{y \in \mathcal{S}}\left(\hat{P}(y|x, a) \cdot \min\left\{\frac{84^2H^3S^2AL^2}{\max\{1, N'_{k,h+1}(y)\}}, H^2\right\}\right)}{\max\{N_k(x, a), 1\}}}}_{(C)},$$

*then, w.p. at least $1 - \delta$, the regret of `UCBVI-BF-I` is*

---

[5]We assume that, by definition, $b_{k,H}(s, a) = 0$, as at the last stage there is no need for exploration and the rewards are deterministic.

[6]To the best of the authors' knowledge, the original analysis of (Azar et al., 2017) is missing a multiplicative $e$ factor in the regret bound.

*bounded by:*

$$\text{Reg}(\texttt{UCBVI-BF-I}, K) \leqslant 24eL\sqrt{HSAT} +$$
$$+ 616eH^2S^2AL^2 + 4e\sqrt{H^2TL},$$

*where* $L = \ln(5HSAT/\delta)$. *For* $T \geqslant H^3S^3A$ *and* $SA \geqslant H$, *this bound translates to* $\tilde{\mathcal{O}}(\sqrt{HSAT})$.

Theorem 3.2 should be compared to Theorem 2 of (Azar et al., 2017). Again, as the analysis is a refinement in terms of constants, the order of the regret does not change. Moreover, also the minimum value of $T$ under which the regret bound holds in unchanged between the two analyses. It is important to notice, however, the strictly smaller constant terms of our analysis w.r.t. the ones of (Azar et al., 2017).[6] In term (A) we show a multiplicative factor of $\sqrt{4}$ instead of $\sqrt{8}$, in term (B) we have a multiplicative term 7 instead of 14, and in term (C) we have a $\sqrt{4}$ multiplicative factor instead of $\sqrt{8}$ and a multiplicative term $84^2$ inside the minimum instead of $100^2$. Such a reduction in the constant values directly affects the behavior of the algorithm by reducing the confidence intervals and reducing, in turn, the unnecessary exploration.

The reader shall refer to Appendices E and F for the proofs of Theorems 3.1 and 3.2, respectively. The derivations provided in the appendices closely follow the proofs of (Azar et al., 2017), focusing on lowering the constant terms. A full description of the notation employed throughout the paper is reported in Appendix B. Both proofs are conducted under the condition that concentration inequalities hold for the next state estimator and its variance. Those conditions fall under event $\mathcal{E}$, which is presented in Appendix B.4 of (Azar et al., 2017) and restated in Appendix C. Finally, additional lemmas necessary to show the regret decomposition and to bound the summation of the terms it comprises are demonstrated in Appendix D.

## 4. Numerical Validation

In this section, we numerically compare the performances of $\texttt{UCBVI}$, both with the Chernoff-Hoeffding and Bernstein-Freedman bonuses of (Azar et al., 2017) and with the improved Bernstein-Freedman bonus of this paper, against the $\texttt{MVP}$ algorithm.[7]

In order to fairly compare to the $\texttt{MVP}$ algorithm, all the $N_{k,h}(x,a)$ terms are considered as $N_k(x,a)$, removing the discriminant of the stage from the algorithm, and the $c_2$ constant (which refers to the uncertainty in the estimation of the rewards) is set to 0, to remove the exploration factor needed due to the stochasticity of the reward in the original

paper. The resulting exploration bound is:

$$b_{k,h}^{\texttt{MVP}}(x,a) = \frac{460}{9}\sqrt{\frac{\mathbb{V}\text{ar}_{y\sim\hat{P}_k(\cdot|x,a)}(V_{k,h+1}(y))\log\frac{1}{\delta}}{\max\{N(x,a),1\}}} +$$
$$+ \frac{544}{9}\frac{H\log\frac{1}{\delta}}{\max\{N(x,a),1\}}.$$

### 4.1. Illustrative Environments

As a first experimental evaluation, we consider a set of illustrative environments. We consider an MDP with parameters $S = 3$, $A = 3$, $H \in \{5, 10\}$, and we consider a number of episodes $K \in \{10^5, 10^6\}$.

We evaluate each experiment by averaging over 10 runs. In each run, the rewards and transition probabilities of the MDP are randomly generated. Then, the clairvoyant optimum is calculated for the purpose of regret computation, and the algorithms are evaluated.

**Results.** Figure 1 shows the cumulative regret of the evaluated algorithms in the first experimental evaluation for different values of $H$ and $K$. From these results, we can observe that $\texttt{UCBVI}$ with the Chernoff-Hoeffding bonus and $\texttt{MVP}$ begin to show a sub-linear regret for $K = 10^6$, whereas both versions of $\texttt{UCBVI}$ with the Bernstein-Freedman bonus greatly outperform the other algorithms in all the evaluated scenarios. In particular, the use of a tighter Bernstein-Freedman bonus ($\texttt{UCBVI-BF-I}$) translates into a cumulative regret that is, although of the same order, lower than with the usage of a larger bonus ($\texttt{UCBVI-BF}$), highlighting the fundamental importance of lower order terms and constants in empirical performance. A further discussion is postponed to Section 5.[8]

### 4.2. RiverSwim

We now consider the RiverSwim environment (Strehl & Littman, 2008). This environment emulates a swimmer that has to swim against the current, where the agent has 2 options: try to swim to the other side or turn back. In this scenario, the rewards and the transition probabilities are designed such that the optimal policy corresponds to trying to swim and reach the other side of the "river". This is considered a challenging benchmark for exploration. We consider the scenario with $S = 5$ and $H = 10$. The reward model and the transition probability are designed such that the suboptimality gap between the optimal action and the other one in the initial state is very low ($\sim 0.1$, with a scale of the problem in the order of $H = 10$).

**Results.** Figure 2 compares the results when using $\texttt{MVP}$ and $\texttt{UCBVI}$ in its original version ($\texttt{UCBVI-BF}$) and the

---

[7]The code to reproduce the experiments can be found at: https://github.com/marcomussi/position_constants.

[8]In Appendix G, we provide additional results on environments with larger state and action spaces.

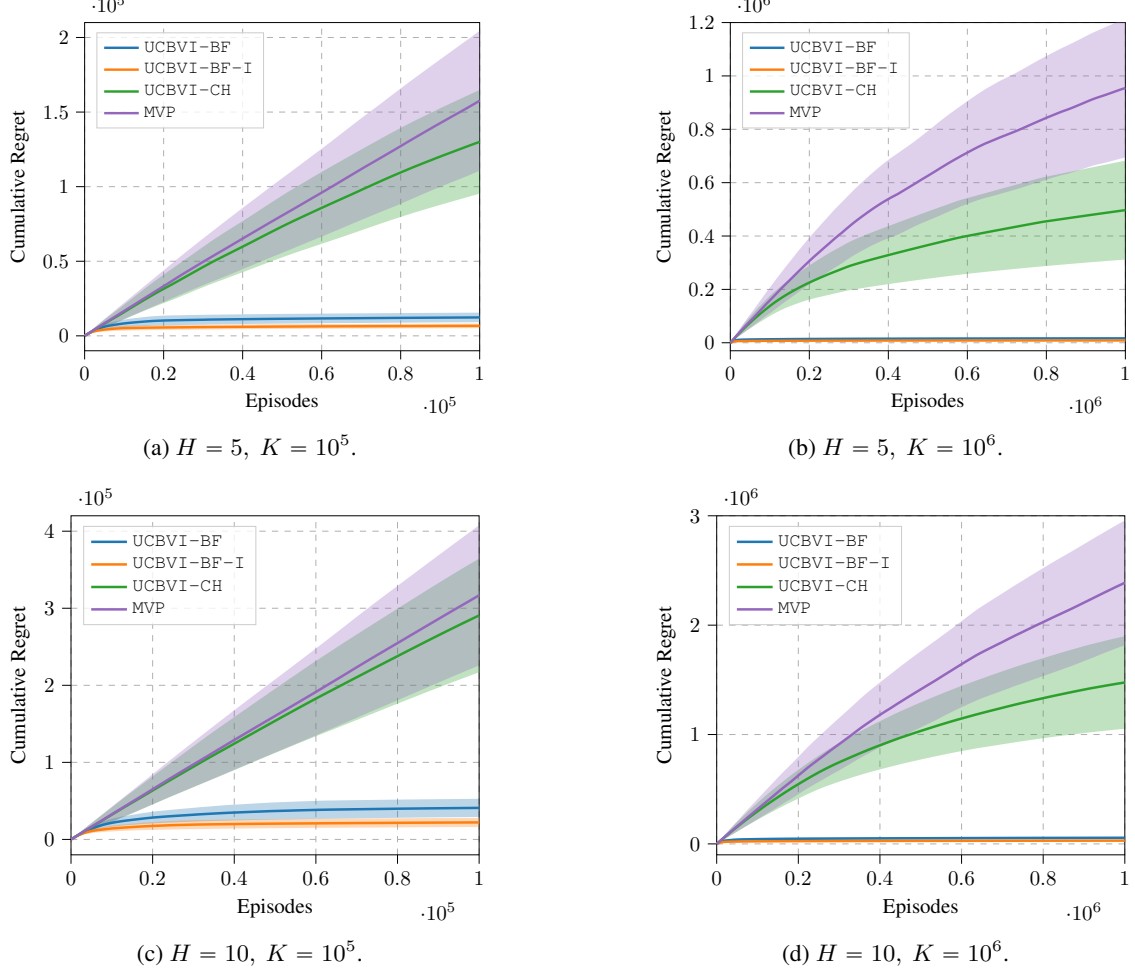

Figure 1. Performances in terms of cumulative regret in toy environments with $S = 3$ states and $A = 3$ actions (10 runs, mean $\pm$ 95% C.I.).

one we propose with tighter bounds (UCBVI-BF-I). MVP confirms its poor empirical performance, failing to deliver a sublinear trend for the considered horizon. Instead, UCBVI, in both versions, shows a clear sublinear trend, with the improved version (UCBVI-BF-I) showing a cumulative regret approximately half of the original one (UCBVI-BF).

## 5. Discussion

In this section, we discuss the results we have obtained from both theoretical and empirical standpoints. A summary of the improvements, expressed in terms of improvement ratios in the bonuses, regret upper bounds, and empirical regret, is reported in Table 1. First, we compare our versions of the UCBVI algorithms with the original ones from (Azar et al., 2017). The algorithmic structure remains the same, though we re-derived the bonus terms to make them as tight as possible, resulting in an improvement of $7/2$ and $\sqrt{2}$ for the Chernoff-Hoeffding and Bernstein-Freedman bonuses,

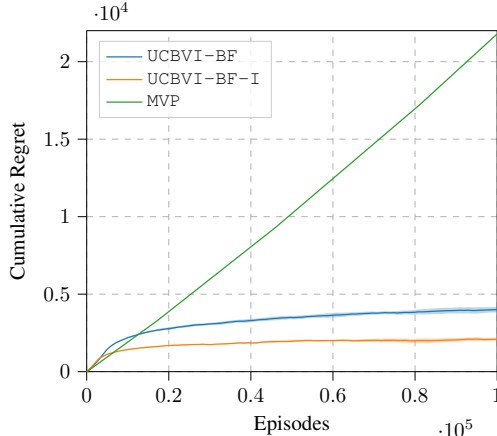

Figure 2. Performances in terms of cumulative regret in the River-Swim environment with $S = 5$ states and horizon $H = 10$ (4 runs, mean $\pm$ 95% C.I.).

|     | Bonus ratio | Regret upper bound ratio | Empirical regret ratio |
| --- | --- | --- | --- |
| CH | $7/2$ | $2$ | - |
| BF | $\sqrt{2}$ | $5/4$ | $1.87 \pm 0.03$ |

*Table 1.* Improvement ratios in bonuses, regret bounds, and empirical regret between our analysis and the original ([Azar et al., 2017]).

in the dominant terms, respectively. This reduction in over-exploration has significant empirical effects, as shown in the experiments in Section 4, where, as reported in Table 1, we achieve a reduction in the empirical regret by a factor of $1.87$. Additionally, this impacts the regret analysis, where we were able to reduce the regret bound by a factor of $2$ and $5/4$ for the Chernoff-Hoeffding and Bernstein-Freedman bonuses, respectively, w.r.t. dominant terms. However, lower order terms also have an impact on the performance, and through a refined analysis, we were able to reduce them by a factor of $\sim 90$ and $\sim 4$ for the Chernoff-Hoeffding and Bernstein-Freedman bonuses, respectively.

Indeed, it is important to highlight that the analyses of these algorithms rely on a well-established set of tools, primarily represented by concentration bounds for martingales. This is particularly evident in the algorithmic approach of MVP ([Zhang et al., 2024]). The algorithm employs a *doubling trick*, which is known for its sample inefficiency, as it requires discarding previously collected samples, albeit at the cost of only a multiplicative constant in the final regret ([Besson & Kaufmann, 2018]). This seems to contradict the intuition commonly held in machine learning that *the more samples, the better*, raising the question of whether the current probabilistic tools are strong enough to effectively capture the properties of the estimators involved.

As a first step towards mitigating this issue, algorithm designers should incorporate not only upper bounds but also *fixed-algorithm regret lower bounds*. These lower bounds illustrate the *minimum* regret that the algorithm can incur under the most challenging scenarios. This approach provides valuable insight into the tightness of both the algorithm and its analysis, enabling a deeper understanding of its performances and limitations.

Moreover, we conjecture that, if the performance difference ascribable to tighter constants and lower order terms is considerable in tabular RL environments, it will be exacerbated when considering more complex algorithms, such as deep RL ones. We believe that, with the necessary mathematical tools and due precautions, it could be possible to apply refinements similar to those we discussed in this paper to such complex settings. Two possible directions to achieve such a result, in the authors' opinion, are $(i)$ to try to study the theoretical guarantees of existing algorithms to gain a deeper knowledge of the settings and in turn allow the definition of better-performing algorithms and $(ii)$ to research

novel mathematical tools, which may enable the analysis of complex settings.

## 6. Alternative Views

In contrast with the perspectives presented in this work, some alternative views may exist.

Firstly, one could argue that the significance of an algorithm, such as MVP, designed to achieve certain theoretical guarantees, extends beyond the tightness of those guarantees (especially regarding constants). Such algorithms may introduce $(i)$ novel technical tools (e.g., concentration inequalities) and $(ii)$ innovative algorithmic solutions, which could hold independent interest.

Secondly, it may be argued that pursuing algorithms with tight theoretical guarantees should not be regarded as the optimal path for advancing this field. Worst-case guarantees are often overly conservative, focusing on ensuring performance in pathological problem instances that rarely occur in practice. Consequently, a substantial portion of the scientific community is shifting toward prioritizing the empirical performance of algorithms to effectively address real-world problems. These algorithms, while potentially lacking general theoretical guarantees, often perform as intended in practical problems.

## 7. Conclusions

In conclusion, the objective of this paper is not to undervalue the research efforts aimed at reducing the regret order of state-of-the-art algorithms. Instead, it seeks to highlight the importance of considering lower-order terms and constants when transitioning algorithms from theoretical frameworks to experimental settings. We hope this position paper serves to reduce the gap between theoretical guarantees and real-world performance, leading to a more integrated view within the RL community.

## Impact Statement

This position paper explores the current advancements and future directions in the field of Machine Learning (ML). Our work aims to provide an analysis of emerging trends, challenges, and opportunities in ML. There are many potential societal consequences of our work, none which we feel must be specifically highlighted here.

## Acknowledgments

Funded by the European Union – Next Generation EU within the project NRPP M4C2, Investment 1.3 DD. 341 – 15 March 2022 – FAIR – Future Artificial Intelligence Research – Spoke 4 – PE00000013 – D53C22002380006.

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

# A. Additional Examples in the Bandit Literature

In this section, we present and discuss examples from the MAB literature of works that, by focusing on deriving tighter constants and lower-order terms, achieve significant performance improvements.

**Multi-Armed Bandits.** In the MAB setting, an agent selects an arm to pull and receives a scalar reward as feedback. (Auer et al., 2002) propose the well-known `UCB1` algorithm, which adopts the *optimism in the face of uncertainty* principle to guide the exploration of the agent. The original version of `UCB1` derives its confidence bound via an argument based on Boole's inequality, later using such a confidence bound to derive its regret bound. Later, (Bubeck, 2010) develop an improved version of the algorithm and, through a more convoluted analysis, improve the confidence bound by a factor $4$ thanks to arguments based on Martingales and to a Peeling argument. As a result, the confidence bound of (Auer et al., 2002) contains a $\sqrt{2\log(T)/n}$ term, where $T$ is the learning horizon and $n$ is the number of times an arm has been pulled, whereas the confidence bound of (Bubeck, 2010) contains a $\sqrt{\alpha \log(t)/n}$ term, where $t$ is the current round, and provides a regret bound for any $\alpha > 1/2$. This modification does not change the order of the regret, which is equal for both the minimax and instance-dependent perspectives (up to constants). However, this refinement has the paramount effect of reducing unnecessary exploration, thus significantly improving the empirical performance of the algorithm. In support of this argument, in Figure 3 we show a comparison between the two versions of the `UCB1` algorithm, for a varying number of actions $A \in \{3, 5, 10\}$ and time horizon $T \in \{10^4, 5 \cdot 10^4, 10^5\}$. This simple experiment demonstrates the impact that improvements can have on the empirical performance of an algorithm, even if they cannot be observed in the order of the regret. The two versions of the `UCB1` algorithm discussed above work in the settings in which the reward is drawn from a subgaussian distribution.

Additionally, in the specific case in which the reward is sampled from a Bernoulli distribution, (Garivier & Cappé, 2011) further improves the performance achievable by an agent. The authors propose the `KL-UCB` algorithm, an optimistic algorithm that employs the Kullback-Leibler (KL) divergence to compute the upper confidence bound of each arm at each round. Through a tight analysis, the authors derive an asymptotically optimal regret bound. Moreover, the authors demonstrate the superior empirical performance of their algorithm against several baselines, among which the original version of `UCB1`, in different scenarios.

**Linear Bandits.** Abbasi-Yadkori et al. (2011) study Linear Bandits, i.e., the setting in which the agent selects an action $X_t \in \mathbb{R}^d$ and receives a reward $Y_t = \langle X_t, \theta_* \rangle + \eta_t$, where $\theta_* \in \mathbb{R}^d$ is an unknown parameter that the agent wants to estimate, and $\eta_t$ is a zero-mean random noise. The authors improve on the work of Dani et al. (2008), modifying it to employ a novel confidence set, which reduces the regret bound of a $\sqrt{\log(T)}$ multiplicative factor, where $T$ is the learning horizon. Although improving the analysis by a lower-order term, the authors then show that the empirical improvement is far more significant, thus demonstrating that modifications affecting lower-order terms in the regret analysis can have a meaningful impact on the practical performance of an algorithm.

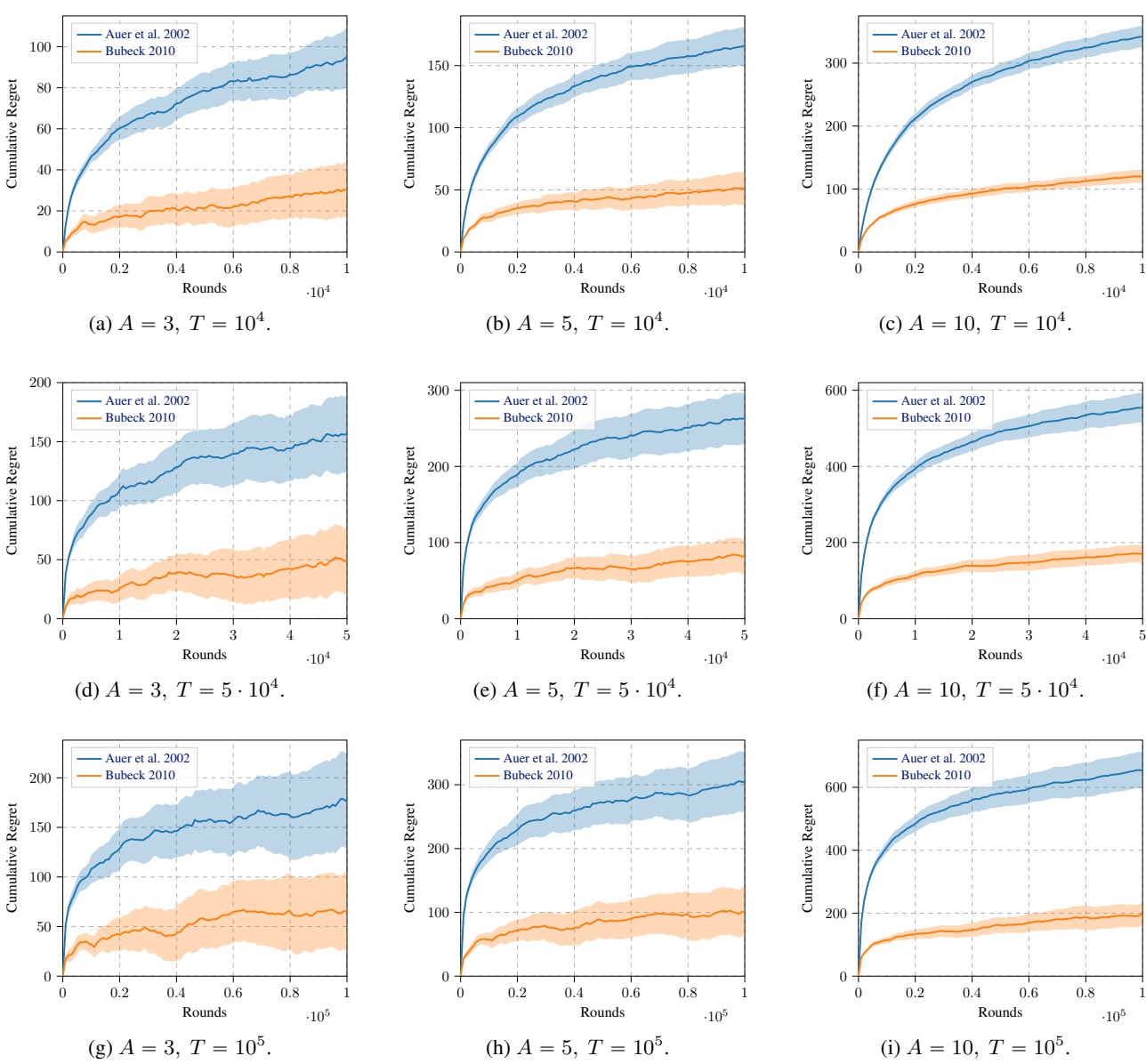

Figure 3. Performances in terms of cumulative regret for a stochastic bandit problem with $A \in \{3, 5, 10\}$ for $T \in \{10^4, 5 \cdot 10^4, 10^5\}$ (50 runs, mean $\pm$ 95% C.I.).

# B. Notation

In this section, we collect the notation used throughout the main paper and the appendices with the related meaning.

| Symbol | Meaning |
|---|---|
| $\mathcal{S}$ | State space |
| $\mathcal{A}$ | Action space |
| $P$ | Transition distribution |
| $R$ | Reward function |
| $H$ | Length of the episode |
| $K$ | Total number of episodes |
| $T$ | Total number of steps |
| $T_k$ | Total number of steps up to episode $k$ |
| $S$ | Cardinality of the state space |
| $A$ | Cardinality of the action space |
| $x_{k,h}$ | State occupied at stage $h$ of episode $k$ |
| $a_{k,h}^{\pi}$ | Action played at stage $h$ of episode $k$ under policy $\pi$ |
| $R^{\pi}(x)$ | Reward obtained by playing according to policy $\pi$ in state $x$ |
| $N_k(x,a)$ | Number of visits to state-action pair $(x,a)$ up to episode $k$ |
| $N_k(x,a,y)$ | Number of transitions to state $y$ from state $x$ after playing action $a$, up to episode $k$ |
| $N'_{k,h}(x)$ | Number of visits to state $x$ at stage $h$ up to episode $k$ |
| $\widehat{P}_k$ | Estimated transition distribution |
| $b_{k,h}$ | Exploration bonus |
| $b'_{k,h}(x)$ | $\min\{\frac{84^2 H^3 S^2 A L^2}{N'_{k,h}(x)}, H^2\}$ |
| $\pi_k$ | Policy played during episode $k$ |
| $\pi^*$ | Optimal policy |
| $Q_{k,h}$ | Optimistic state-action value function |
| $V_h^*$ | Value function of the optimal policy at stage $h$ |
| $V_h^{\pi}$ | Value function under policy $\pi$ at stage $h$ |
| $V_{k,h}$ | Optimistic estimator of the optimal value function at stage $h$ of episode $k$ |
| $\Delta_{k,h}(x)$ | Regret in state $x$, at stage $h$ of episode $k$, following policy $\pi_k$ |
| $\widetilde{\Delta}_{k,h}(x)$ | Pseudo-regret in state $x$, at stage $h$ of episode $k$, following policy $\pi_k$ |
| $\mathrm{Reg}(\mathtt{UCBVI-CH}, k)$ | Regret of $\mathtt{UCBVI}$ using Chernoff-Hoeffding bonus after $k$ episodes |
| $\widetilde{\mathrm{Reg}}(\mathtt{UCBVI-CH}, k)$ | Pseudo-regret of $\mathtt{UCBVI}$ using Chernoff-Hoeffding bonus after $k$ episodes |
| $\mathrm{Reg}(\mathtt{UCBVI-BF-I}, k)$ | Regret of $\mathtt{UCBVI}$ using Bernstein-Freedman bonus after $k$ episodes |
| $\widetilde{\mathrm{Reg}}(\mathtt{UCBVI-BF-I}, k)$ | Pseudo-regret of $\mathtt{UCBVI}$ using Bernstein-Freedman bonus after $k$ episodes |
| $\mathcal{E}$ | Concentration inequalities event |
| $\Omega, \Omega_{k,h}$ | Optimism events |
| $\varepsilon, \overline{\varepsilon}$ | Martingale differences sequences |
| $[k]_{\mathrm{typ}}, [k]_{\mathrm{typ},x}$ | Sets of typical episodes |
| $\mathcal{H}_{k,h}$ | History of the interactions up to, and including, stage $h$ of episode $k$ |
| $L$ | Logarithmic term $\ln(5HSAT/\delta)$ |
| $\mathbb{V}_h^{\pi_k}(x,a)$ | Next-state variance of $V^{\pi_k}$ |
| $\mathbb{V}_h^*$ | Next-state variance of $V^*$ |
| $\widehat{\mathbb{V}}_{k,h}$ | Empirical next-state variance of $V_{k,h}$ |
| $\widehat{\mathbb{V}}_{k,h}^*$ | Empirical next-state variance of $V^*$ |
| $\xi_{k,j}(x,a)$ | State-action wise model error $\xi_{k,j}(x,a) := \sum_{y\in\mathcal{S}}[\widehat{P}_k(y\vert x,a) - P(y\vert x,a)]V_{h+1}^*(y)$ |

*Table 2.* Table of notation.

We define the empirical next state variance of $V$ as:

$$\widehat{\mathbb{V}}_{k,h+1}(x,a) := \underset{y \sim \widehat{P}_k(\cdot|x,a)}{\mathbb{V}\text{ar}} [V_{k,h+1}(y)].$$

We define the next state variance of $V^*$ as:

$$\mathbb{V}^*_{h+1}(x,a) := \underset{y \sim P(\cdot|x,a)}{\mathbb{V}\text{ar}} [V^*_{h+1}(y)].$$

We define the next state variance of $V^\pi$ as:

$$\mathbb{V}^\pi_{h+1}(x,a) := \underset{y \sim P(\cdot|x,a)}{\mathbb{V}\text{ar}} [V^\pi_{h+1}(y)].$$

Finally, we define the empirical next state variance of $V^*$ as:

$$\widehat{\mathbb{V}}^*_{k,h+1}(x,a) := \underset{y \sim \widehat{P}_k(\cdot|x,a)}{\mathbb{V}\text{ar}} [V^*_{h+1}(y)].$$

## C. High Probability Events

In this section, we restate the high probability event $\mathcal{E}$ under which the concentration inequalities hold, presented in Appendix B.4 of (Azar et al., 2017).

Event $\mathcal{E}$ is defined as:

$$\mathcal{E} := \mathcal{E}_{\widehat{P}} \bigcap \bigcap_{\substack{k \in [\![K]\!] \\ h \in [\![H]\!] \\ x \in \mathcal{S}}} \left[ \mathcal{E}_{\text{az}} \left( \mathcal{F}_{\widetilde{\Delta},k,h}, H, L \right) \bigcap \mathcal{E}_{\text{az}} \left( \mathcal{F}'_{\widetilde{\Delta},k,h}, \frac{1}{\sqrt{L}}, L \right) \bigcap \mathcal{E}_{\text{az}} \left( \mathcal{F}_{\widetilde{\Delta},k,h,x}, H, L \right) \right.$$

$$\bigcap \mathcal{E}_{\text{az}} \left( \mathcal{F}'_{\widetilde{\Delta},k,h,x}, \frac{1}{\sqrt{L}}, L \right) \bigcap \mathcal{E}_{\text{fr}} \left( \mathcal{G}_{\mathbb{V},k,h}, H^4 T, H^3, L \right)$$

$$\bigcap \mathcal{E}_{\text{fr}} \left( \mathcal{G}_{\mathbb{V},k,h,x}, H^5 N'_{k,h}(x), H^3, L \right) \bigcap \mathcal{E}_{\text{az}} \left( \mathcal{F}_{b',k,h}, H^2, L \right)$$

$$\left. \bigcap \mathcal{E}_{\text{az}} \left( \mathcal{F}_{b',k,h,x}, H^2, L \right) \right]$$

We refer the reader to Lemma 1 of (Azar et al., 2017) for the proof that event $\mathcal{E}$ holds with high probability. Let, for ease of reading $\overline{x} = x_{i,j}$, $\overline{x}' = x_{i,j+1}$, and $\overline{a} = a^{\pi_i}_{i,j}$ We now restate the definition of the events that compose $\mathcal{E}$:

$$\mathcal{E}_{\widehat{P}} := \left\{ \widehat{P}_k(y|x,a) \in \mathcal{P}(k,h,N_k(x,a),x,a,y), \forall k \in [\![K]\!], h \in [\![H]\!], (x,a,y) \in \mathcal{S} \times \mathcal{A} \times \mathcal{S} \right\},$$

where $\mathcal{P}(k,h,n,x,a,y)$ is defined as the subset of the set of all probability distributions $\mathcal{P}$ over $\mathcal{S}$ such that:

$$\mathcal{P}(k,h,n,x,a,y) := \left\{ \widetilde{P}(\cdot|x,a) \in \mathcal{P} : \|\widetilde{P}(\cdot|x,a) - P(\cdot|x,a)\|_1 \leqslant 2\sqrt{\frac{SL}{n}}, \right. \tag{1}$$

$$\left| \sum_{y \in \mathcal{S}} (\widetilde{P}(y|x,a) - P(y|x,a))V_h^*(y) \right|$$

$$\leqslant \min\left( \sqrt{\frac{2\widehat{\mathbb{V}}_{k,h+1}^*(x,a)L}{n}} + \frac{7HL}{3(n-1)}, \sqrt{\frac{2\mathbb{V}_{h+1}^*(x,a)L}{n}} + \frac{2HL}{3n} \right) \tag{2}$$

$$\left. |\widetilde{P}(y|x,a) - P(y|x,a)| \leqslant \sqrt{\frac{2p(1-p)L}{n}} + \frac{2L}{3n}, \right\}, \tag{3}$$

where Equation (1) follows by applying the result of Theorem 2.1 of (Weissman et al., 2003), Equation (2) follows by applying both Bernstein's inequality (see, e.g., Cesa-Bianchi & Lugosi, 2006) and the empirical Bernstein inequality (Maurer & Pontil, 2009) ,and Equation (3) follows by applying Lemma D.1.

$$\mathcal{E}_{\mathrm{az}}\left(\mathcal{F}_{\widetilde{\Delta},k,h}, H, L\right) := \left\{ \sum_{i=1}^{k} \sum_{j=h}^{H-1} \left[ \sum_{y \in \mathcal{S}} P(y|\overline{x},\overline{a})\widetilde{\Delta}_{i,j+1}(y) - \widetilde{\Delta}_{i,j+1}(\overline{x}') \right] \leqslant 2\sqrt{k(H-h)H^2L} \right\},$$

$$\mathcal{E}_{\mathrm{az}}\left(\mathcal{F}'_{\widetilde{\Delta},k,h}, \frac{1}{\sqrt{L}}, L\right) := \left\{ \sum_{i=1}^{k} \sum_{j=h}^{H} \left[ \sum_{y \in \mathcal{S}} P(y|\overline{x},\overline{a})\sqrt{\frac{\mathbb{I}(y \in [y]_{i,j})}{N_i(\overline{x},\overline{a})P(y|\overline{x},\overline{a})}}\widetilde{\Delta}_{i,j+1}(y) \right] \right.$$
$$\left. - \sqrt{\frac{\mathbb{I}(y \in [y]_{i,j})}{N_i(\overline{x},\overline{a})P(y|\overline{x},\overline{a})}}\widetilde{\Delta}_{i,j+1}(\overline{x}') \leqslant 2\sqrt{k(H-h)\frac{1}{\sqrt{L}^2}L} \right\},$$

$$\mathcal{E}_{\mathrm{az}}\left(\mathcal{F}_{\widetilde{\Delta},k,h,x}, H, L\right) := \left\{ \sum_{i=1}^{k} \mathbb{I}(x_{i,h} = x) \sum_{j=h}^{H-1} \left[ \sum_{y \in \mathcal{S}} P(y|\overline{x},\overline{a})\widetilde{\Delta}_{i,j+1}(y) - \widetilde{\Delta}_{i,j+1}(\overline{x}') \right] \leqslant 2\sqrt{N'_{k,h}(x)(H-h)H^2L} \right\},$$

$$\mathcal{E}_{\mathrm{az}}\left(\mathcal{F}'_{\widetilde{\Delta},k,h,x}, \frac{1}{\sqrt{L}}, L\right) := \left\{ \sum_{i=1}^{k} \mathbb{I}(x_{i,h} = x)\left[ \sum_{j=h}^{H} \left[ \sum_{y \in \mathcal{S}} P(y|\overline{x},\overline{a})\sqrt{\frac{\mathbb{I}(y \in [y]_{i,j})}{N_i(\overline{x},\overline{a})P(y|\overline{x},\overline{a})}}\widetilde{\Delta}_{i,j+1}(y) \right] \right. \right.$$
$$\left. \left. - \sqrt{\frac{\mathbb{I}(y \in [y]_{i,j})}{N_i(\overline{x},\overline{a})P(y|\overline{x},\overline{a})}}\widetilde{\Delta}_{i,j+1}(\overline{x}') \right] \leqslant 2\sqrt{N'_{k,h}(x)(H-h)\frac{1}{\sqrt{L}^2}L} \right\},$$

$$\mathcal{E}_{\mathrm{fr}}\left(\mathcal{G}_{\mathbb{V},k,h}, H^4T, H^3, L\right) := \left\{ \sum_{i=1}^{k} \mathbb{E}\left[ \sum_{j=h}^{H-1} \mathbb{V}_{j+1}^{\pi_k}(\overline{x},\overline{a})|\mathcal{H}_{k,h} \right] - \sum_{i=1}^{k} \sum_{j=h}^{H-1} \mathbb{V}_{j+1}^{\pi_k}(\overline{x},\overline{a}) \right\}$$
$$\leqslant 2\sqrt{H^4T_kL} + \frac{4H^3L}{3},$$

$$\mathcal{E}_{\mathrm{fr}}\left(\mathcal{G}_{\mathbb{V},k,h,x}, H^5N'_{k,h}(x), H^3, L\right) := \left\{ \sum_{i=1}^{k} \mathbb{I}(x_{i,h} = x)\mathbb{E}\left[ \sum_{j=h}^{H-1} \mathbb{V}_{j+1}^{\pi_k}(\overline{x},\overline{a})|\mathcal{H}_{k,h} \right] - \sum_{i=1}^{k} \mathbb{I}(x_{i,h} = x) \sum_{j=h}^{H-1} \mathbb{V}_{j+1}^{\pi_k}(\overline{x},\overline{a}) \right\}$$
$$\leqslant 2\sqrt{H^5N'_{k,h}(x)L} + \frac{4H^3L}{3},$$

$$\mathcal{E}_{\text{az}}\left(\mathcal{F}_{b',k,h}, H^2, L\right) := \left\{ \sum_{i=1}^{k} \sum_{j=h}^{H-1} \sum_{y \in \mathcal{S}} P(y|\overline{x}, \overline{a}) b'_{i,j+1}(y) - b'_{i,j+1}(\overline{x}') \right\}$$

$$\leqslant 2\sqrt{k(H-h)H^4 L}$$

$$\mathcal{E}_{\text{az}}\left(\mathcal{F}_{b',k,h,x}, H^2, L\right) := \left\{ \sum_{i=1}^{k} \mathbb{I}(x_{i,h} = x) \left[ \sum_{j=h}^{H-1} \sum_{y \in \mathcal{S}} P(y|\overline{x}, \overline{a}) b'_{i,j+1}(y) - b'_{i,j+1}(\overline{x}') \right] \right\}$$

$$\leqslant 2\sqrt{N'_{k,h}(x)(H-h)H^4 L}$$

## D. Technical Lemmas

**Lemma D.1** (Bernstein inequality for Bernoulli random variables)**.** *Let $p$ be the parameter of a Bernoulli random variable, and let $\widehat{p}$ be its estimator. Let $\delta > 0$. Then, w.p. at least $1 - \delta$, it holds that:*

$$|\widehat{p} - p| \leqslant \sqrt{\frac{2p(1-p)L}{n}} + \frac{2L}{3n},$$

*where $n$ represents the number of observations, and $L = \ln(2/\delta)$.*

*Proof.* Let $\{Y_i\}_{i=1\ldots,n}$ be the set of i.i.d. realizations of a Bernoulli with parameter $p$. Define the auxiliary random variable:

$$X_i = \frac{Y_i}{n}.$$

Observe that $X_1, \ldots, X_n$ are independent random variables, and that $0 \leqslant X_i \leqslant 1/n$. Let $S_n$ be their sum, and $E_n$ be the expected value of $S_n$:

$$S_n = \sum_{i=1}^{n} X_i = \widehat{p},$$

$$E_n = \mathbb{E}[S_n] = \sum_{i=1}^{n} \mathbb{E}[X_i] = p.$$

Let $V_n$ be the variance of $S_n$:

$$V_n = \mathbb{V}\text{ar}[S_n] = \sum_{i=1}^{n} \mathbb{V}\text{ar}[X_i] = \sum_{i=1}^{n} \left(\frac{1(1-p)}{n^2}\right) = \frac{p(1-p)}{n}.$$

By applying Bernstein's inequality, we obtain that:

$$\Pr(|S_n - E_n| > \epsilon) < 2\exp\left(-\frac{\epsilon^2/2}{V_n + C(\epsilon/3)}\right), \tag{4}$$

where $C$ is the range of values of the addends in $S_n$ (i.e., $C = 1/n$). By setting this probability to be equal to $\delta$, we can derive that:

$$\frac{\epsilon^2}{2} = V_n \ln\left(\frac{2}{\delta}\right) + \frac{\epsilon}{3n} \ln\left(\frac{2}{\delta}\right).$$

Let $L = \ln(2/\delta)$, by solving the second order polynomial we get that:

$$\epsilon = \frac{L}{3n} \pm \sqrt{\frac{L^2}{9n^2} + 2V_n L}.$$

We can discard the equation with the minus, as it would result in $\epsilon < 0$, and consequently in the inequality in Equation (4) holding w.p. 1. As such, we derive that:

$$
\begin{aligned}
\epsilon &= \frac{L}{3n} + \sqrt{\frac{L^2}{9n^2} + 2V_n L} \\
&\leqslant \frac{L}{3n} + \sqrt{\frac{L^2}{9n^2}} + \sqrt{\frac{2p(1-p)L}{n}} \\
&= \sqrt{\frac{2p(1-p)L}{n}} + \frac{2L}{3n},
\end{aligned}
$$

thus completing the proof. $\square$

**Lemma D.2** (Regret decomposition upper bound). *Let $k \in [\![K]\!]$ and $h \in [\![H]\!]$. Assume events $\mathcal{E}$ and $\Omega_{k,h}$ hold. Then the regret from stage $h$ onward of all episodes up to $k$ can be upper bounded as follows:*

$$
\begin{aligned}
\sum_{i=1}^{k} \Delta_{i,h}(x_{i,h}) \leqslant \sum_{i=1}^{k} \widetilde{\Delta}_{i,h}(x_{i,h}) \leqslant e \sum_{i=1}^{k} \sum_{j=h}^{H-1} \Big[ & \varepsilon_{i,j} + 2\sqrt{L}\bar{\varepsilon}_{i,j} + b_{i,j}(x_{i,j}, a_{i,j}^{\pi_i}) \\
& + \xi_{i,j}(x_{i,j}, a_{i,j}^{\pi_i}) + \frac{8H^2 SL}{3N_i(x_{i,j}, a_{i,j}^{\pi_i})} \Big].
\end{aligned}
$$

*Proof.* We begin the proof by considering a single value of $k \in [\![K]\!]$. Under $\Omega_{k,h}$, we observe that:

$$
\begin{aligned}
\Delta_{k,h}(x_{k,h}) &= V_h^*(x_{k,h}) - V_h^{\pi_k}(x_{k,h}) \\
&\leqslant V_{k,h}(x_{k,h}) - V_h^{\pi_k}(x_{k,h}) \\
&= \widetilde{\Delta}_{k,h}(x_{k,h}).
\end{aligned}
$$

As such, we bound the pseudo-regret $\widetilde{\Delta}_{k,h}(x_{k,h})$:

$$\widetilde{\Delta}_{k,h}(x_{k,h}) = V_{k,h}(x_{k,h}) - V_h^{\pi_k}(x_{k,h})$$

$$= b_{k,h}(x_{k,h}, a_{k,h}^{\pi_k}) + \sum_{y \in \mathcal{S}} \widehat{P}_k(y|x_{k,h}, a_{k,h}^{\pi_k}) V_{k,h+1}(y) - \sum_{y \in \mathcal{S}} P(y|x_{k,h}, a_{k,h}^{\pi_k}) V_{h+1}^{\pi_k}(y)$$

$$= b_{k,h}(x_{k,h}, a_{k,h}^{\pi_k}) + \sum_{y \in \mathcal{S}} \left[ \widehat{P}_k(y|x_{k,h}, a_{k,h}^{\pi_k}) - P(y|x_{k,h}, a_{k,h}^{\pi_k}) \right] V_{k,h+1}(y)$$

$$+ \sum_{y \in \mathcal{S}} P(y|x_{k,h}, a_{k,h}^{\pi_k}) \left[ V_{k,h+1}(y) - V_{h+1}^{\pi_k}(y) \right]$$

$$= b_{k,h}(x_{k,h}, a_{k,h}^{\pi_k}) + \sum_{y \in \mathcal{S}} \left[ \widehat{P}_k(y|x_{k,h}, a_{k,h}^{\pi_k}) - P(y|x_{k,h}, a_{k,h}^{\pi_k}) \right] V_{h+1}^*(y)$$

$$+ \sum_{y \in \mathcal{S}} \left[ \widehat{P}_k(y|x_{k,h}, a_{k,h}^{\pi_k}) - P(y|x_{k,h}, a_{k,h}^{\pi_k}) \right] \left[ V_{k,h+1}(y) - V_{h+1}^*(y) \right]$$

$$+ \sum_{y \in \mathcal{S}} P(y|x_{k,h}, a_{k,h}^{\pi_k}) \widetilde{\Delta}_{k,h+1}(y)$$

$$= \widetilde{\Delta}_{k,h+1}(x_{k,h+1}) + b_{k,h}(x_{k,h}, a_{k,h}^{\pi_k}) + \varepsilon_{k,h} \tag{5}$$

$$\sum_{y \in \mathcal{S}} \left[ \widehat{P}_k(y|x_{k,h}, a_{k,h}^{\pi_k}) - P(y|x_{k,h}, a_{k,h}^{\pi_k}) \right] V_{h+1}^*(y)$$

$$+ \sum_{y \in \mathcal{S}} \left[ \widehat{P}_k(y|x_{k,h}, a_{k,h}^{\pi_k}) - P(y|x_{k,h}, a_{k,h}^{\pi_k}) \right] \left[ V_{k,h+1}(y) - V_{h+1}^*(y) \right]$$

$$\leqslant \widetilde{\Delta}_{k,h+1}(x_{k,h+1}) + b_{k,h}(x_{k,h}, a_{k,h}^{\pi_k}) + \xi_{k,h}(x_{k,h}, a_{k,h}^{\pi_k})$$

$$+ \varepsilon_{k,h} + \underbrace{\sum_{y \in \mathcal{S}} \left[ \widehat{P}_k(y|x_{k,h}, a_{k,h}^{\pi_k}) - P(y|x_{k,h}, a_{k,h}^{\pi_k}) \right] \left[ V_{k,h+1}(y) - V_{h+1}^*(y) \right]}_{(a)}, \tag{6}$$

where, in Equation (5) we apply the definition $\varepsilon_{k,h} := P(\cdot|x_{k,h}, a_{k,h}^{\pi_k})^{\mathsf{T}} \widetilde{\Delta}_{k,h+1}(\cdot) - \widetilde{\Delta}_{k,h+1}(x_{k,h+1})$, and in Equation (6) we apply the definition of $\xi_{k,h}(x_{k,h}, a_{k,h}^{\pi_k})$:

$$\xi_{k,h}(x_{k,h}, a_{k,h}^{\pi_k}) = \sum_{y \in \mathcal{S}} \left[ \widehat{P}_k(y|x_{k,h}, a_{k,h}^{\pi_k}) - P(y|x_{k,h}, a_{k,h}^{\pi_k}) \right] V_{h+1}^*(y).$$

Let $\mathcal{H}_{k,h}$ be the history of the interactions up to, and including, stage $h$ of episode $k$. Observing that $|\varepsilon_{k,h}| \leqslant H \leqslant +\infty$ and $\mathbb{E}[\varepsilon_{k,h}|\mathcal{H}_{k,h}] = 0$, we can derive that $\varepsilon_{k,h}$ is a Martingale difference sequence.

We now focus on bounding term $(a)$:

$$(a) = \sum_{y \in \mathcal{S}} \left[ \widehat{P}_k(y|x_{k,h}, a_{k,h}^{\pi_k}) - P(y|x_{k,h}, a_{k,h}^{\pi_k}) \right] \left[ V_{k,h+1}(y) - V_{h+1}^*(y) \right]$$

$$\leqslant \sum_{y \in \mathcal{S}} \left[ \sqrt{\frac{2P(y|x_{k,h}, a_{k,h}^{\pi_k})(1 - P(y|x_{k,h}, a_{k,h}^{\pi_k}))L}{N_k(x_{k,h}, a_{k,h}^{\pi_k})}} + \frac{2L}{3N_k(x_{k,h}, a_{k,h}^{\pi_k})} \right] \widetilde{\Delta}_{k,h+1}(y) \tag{7}$$

$$\leqslant \sum_{y \in \mathcal{S}} \sqrt{\frac{2P(y|x_{k,h}, a_{k,h}^{\pi_k})L}{N_k(x_{k,h}, a_{k,h}^{\pi_k})}} \widetilde{\Delta}_{k,h+1}(y) + \frac{2L}{3N_k(x_{k,h}, a_{k,h}^{\pi_k})} \sum_{y \in \mathcal{S}} \widetilde{\Delta}_{k,h+1}(y) \tag{8}$$

$$\leqslant \sqrt{2L} \underbrace{\sum_{y \in \mathcal{S}} \sqrt{\frac{2P(y|x_{k,h}, a_{k,h}^{\pi_k})L}{N_k(x_{k,h}, a_{k,h}^{\pi_k})}} \widetilde{\Delta}_{k,h+1}(y)}_{(b)} + \frac{2SHL}{3N_k(x_{k,h}, a_{k,h}^{\pi_k})}, \tag{9}$$

where Equation (7) is obtained by applying Lemma D.1 to bound $\widehat{P}_k - P$, Equation (8) is obtained by splitting the terms and observing that $1 - P(y|x, a) \leqslant 1$ for every $x, y \in \mathcal{S}$ and $a \in \mathcal{A}$, and finally Equation (9) is obtained by upper bounding $\widetilde{\Delta}_{k,h+1}(y)$ with $H$. To bound term $(b)$, we first need to define the following set of states:

$$[y]_{k,h} := \{y \in \mathcal{S} : N_k(x_{k,h}, a_{k,h}^{\pi_k})P(y|x_{k,h}, a_{k,h}^{\pi_k}) \geqslant 2H^2L\}.$$

As such, we can rewrite:

$$(b) = \underbrace{\sum_{y \in [y]_{k,h}} \sqrt{\frac{2P(y|x_{k,h}, a_{k,h}^{\pi_k})L}{N_k(x_{k,h}, a_{k,h}^{\pi_k})}} \widetilde{\Delta}_{k,h+1}(y)}_{(c)} + \underbrace{\sum_{y \notin [y]_{k,h}} \sqrt{\frac{2P(y|x_{k,h}, a_{k,h}^{\pi_k})L}{N_k(x_{k,h}, a_{k,h}^{\pi_k})}} \widetilde{\Delta}_{k,h+1}(y)}_{(d)}. \tag{10}$$

We now bound term $(c)$ as:

$$\begin{aligned}
(c) &= \sum_{y \in [y]_{k,h}} \sqrt{\frac{2P(y|x_{k,h}, a_{k,h}^{\pi_k})L}{N_k(x_{k,h}, a_{k,h}^{\pi_k})}} \widetilde{\Delta}_{k,h+1}(y) \\
&= \sum_{y \in [y]_{k,h}} P(y|x_{k,h}, a_{k,h}^{\pi_k})\sqrt{\frac{1}{N_k(x_{k,h}, a_{k,h}^{\pi_k})P(y|x_{k,h}, a_{k,h}^{\pi_k})}} \widetilde{\Delta}_{k,h+1}(y) \\
&= \bar{\varepsilon}_{k,h} + \sqrt{\frac{\mathbb{I}(x_{k,h+1} \in [y]_{k,h})}{N_k(x_{k,h}, a_{k,h}^{\pi_k})P(y|x_{k,h}, a_{k,h}^{\pi_k})}} \widetilde{\Delta}_{k,h+1}(x_{k,h+1}) \tag{11} \\
&\leqslant \bar{\varepsilon}_{k,h} + \sqrt{\frac{1}{2H^2L}} \widetilde{\Delta}_{k,h+1}(x_{k,h+1}), \tag{12}
\end{aligned}$$

where Equation (11) is obtained by applying the definition of $\bar{\varepsilon}_{k,h}$:

$$\begin{aligned}
\bar{\varepsilon}_{k,h} := \sum_{y \in \mathcal{S}} P(y|x_{k,h}, a_{k,h}^{\pi_k})&\sqrt{\frac{\mathbb{I}(y \in [y]_{k,h})}{N_k(x_{k,h}, a_{k,h}^{\pi_k})P(y|x_{k,h}, a_{k,h}^{\pi_k})}} \widetilde{\Delta}_{k,h+1}(y) \\
&- \sqrt{\frac{\mathbb{I}(y \in [y]_{k,h})}{N_k(x_{k,h}, a_{k,h}^{\pi_k})P(y|x_{k,h}, a_{k,h}^{\pi_k})}} \widetilde{\Delta}_{k,h+1}(x_{k,h+1}),
\end{aligned}$$

and Equation (12) is obtained by bounding the indicator function with 1, and by applying the definition of $[y]_{k,h}$. With the same reasoning of $\varepsilon_{k,h}$, we can prove that $\bar{\varepsilon}_{k,h}$ is also a Martingale difference sequence.

We can now bound term $(d)$ as follows:

$$\begin{aligned}
(d) &= \sum_{y \notin [y]_{k,h}} \sqrt{\frac{P(y|x_{k,h}, a_{k,h}^{\pi_k})}{N_k(x_{k,h}, a_{k,h}^{\pi_k})}} \widetilde{\Delta}_{k,h+1}(y) \\
&= \sum_{y \notin [y]_{k,h}} \sqrt{\frac{N_k(x_{k,h}, a_{k,h}^{\pi_k})P(y|x_{k,h}, a_{k,h}^{\pi_k})}{(N_k(x_{k,h}, a_{k,h}^{\pi_k}))^2}} \widetilde{\Delta}_{k,h+1}(y) \\
&\leqslant \frac{H^2 S\sqrt{2L}}{N_k(x_{k,h}, a_{k,h}^{\pi_k})}, \tag{13}
\end{aligned}$$

where Equation (13) is obtained by bounding $\widetilde{\Delta}_{k,h+1}(y)$ with $H$, and by applying the definition of $[y]_{k,h}$. We can now plug the bounds of $(c)$ and $(d)$ into Equation (10) to obtain that:

$$(b) \leqslant \overline{\varepsilon}_{k,h} + \sqrt{\frac{1}{2H^2L}} \widetilde{\Delta}_{k,h+1}(x_{k,h+1}) + \frac{H^2 S \sqrt{2L}}{N_k(x_{k,h}, a_{k,h}^{\pi_k})}.$$

By plugging the bound of $(b)$ into Equation (9), we obtain that:

$$(a) \leqslant \sqrt{2L}\overline{\varepsilon}_{k,h} + \frac{1}{H} \widetilde{\Delta}_{k,h+1}(x_{k,h+1}) + \frac{8H^2 SL}{3N_k(x_{k,h}, a_{k,h}^{\pi_k})}.$$

Finally, substituting the bound on $(a)$ into Equation (6), we obtain that:

$$\widetilde{\Delta}_{k,h}(x_{k,h}) \leqslant \left(1 + \frac{1}{H}\right) \widetilde{\Delta}_{k,h+1}(x_{k,h+1}) + b_{k,h}(x_{k,h}, a_{k,h}^{\pi_k}) + \xi_{k,h}(x_{k,h}, a_{k,h}^{\pi_k}) + \varepsilon_{k,h} + \sqrt{2L}\overline{\varepsilon}_{k,h} + \frac{8H^2 SL}{3N_k(x_{k,h}, a_{k,h}^{\pi_k})}.$$

We now apply an inductive argument on $\widetilde{\Delta}_{k,h}(x_{k,h})$ to isolate the term.

Observing that $\widetilde{\Delta}_{k,H+1}(x_{k,H+1}) = 0$ by definition, we can rewrite:

$$\widetilde{\Delta}_{k,h}(x_{k,h}) \leqslant \sum_{j=h}^{H-1} \gamma_{j-h} \left[ b_{k,j}(x_{k,j}, a_{k,j}^{\pi_k}) + \xi_{k,j}(x_{k,j}, a_{k,j}^{\pi_k}) + \varepsilon_{k,j} + \sqrt{2L}\overline{\varepsilon}_{k,j} + \frac{8H^2 SL}{3N_k(x_{k,j}, a_{k,j}^{\pi_k})} \right],$$

where $\gamma_{j-h} = \left(1 + \frac{1}{H}\right)^{j-h}$. Notice that the summation is limited to $H-1$. This will be recurrent throughout the paper and is due to the fact that, the reward being deterministic, there is no uncertainty at $h = H$. As such, we can assume that the policies $\pi_k$ for $k \in [\![K]\!]$ always play greedily at the last stage of each episode.

Observing that $1 + \frac{1}{H} > 1$, we trivially derive that $\gamma_{j-h} \leqslant \gamma_H$ for $j \in [\![h, H]\!]$. Recalling that $\lim_{x \to +\infty} \left(1 + \frac{1}{x}\right)^x = e$, we can bound $\gamma_H \leqslant e$, and rewrite:

$$\widetilde{\Delta}_{k,h}(x_{k,h}) \leqslant e \sum_{j=h}^{H-1} \left[ b_{k,j}(x_{k,j}, a_{k,j}^{\pi_k}) + \xi_{k,j}(x_{k,j}, a_{k,j}^{\pi_k}) + \varepsilon_{k,j} + \sqrt{2L}\overline{\varepsilon}_{k,j} + \frac{8H^2 SL}{3N_k(x_{k,j}, a_{k,j}^{\pi_k})} \right], \tag{14}$$

To conclude the proof, we need now to show that this holds for any value of $k \in [\![K]\!]$. Recalling the definition of $\Omega_{k,h}$:

$$\Omega_{k,h} := \left\{ V_{i,j}(x) \geqslant V_j^*(x), \forall (i,j) \in [k,h]_{\text{hist}}, x \in \mathcal{S} \right\},$$

where $[k,h]_{\text{hist}} := \{(i,j) : i \in [\![K]\!], j \in [\![H]\!], (i < k) \vee (i = k, j \geqslant h)\}$, we observe that, if $\Omega_{k,h}$ holds, then also the events $\Omega_{i,j}$ hold for $(i,j) \in [k,h]_{hist}$. As such, we can sum up the previous bound of Equation (14) over all the episodes $i \in [\![k]\!]$, thus concluding the proof. $\qquad\square$

**Lemma D.3.** *Let $k \in [\![K]\!]$ and $h \in [\![H]\!]$. Let events $\mathcal{E}$ and $\Omega_{k,h}$ hold. Then the following bounds hold:*

$$\sum_{i=1}^{k} \sum_{j=h}^{H} \varepsilon_{i,j} \leqslant 2\sqrt{H^2 T_k L},$$

$$\sum_{i=1}^{k} \sum_{j=h}^{H} \overline{\varepsilon}_{i,j} \leqslant 2\sqrt{T_k},$$

*where $T_k = kH$.*

*Proof.* Let us first recall the definitions of $\varepsilon_{i,j}$ and $\overline{\varepsilon}_{i,j}$:

$$\varepsilon_{i,j} := P(\cdot|x_{i,j}, a_{i,j}^{\pi_i})^{\mathsf{T}} \widetilde{\Delta}_{i,j+1}(\cdot) - \widetilde{\Delta}_{i,j+1}(x_{i,j+1}),$$

$$\overline{\varepsilon}_{i,j} := \sum_{y \in \mathcal{S}} P(y|x_{i,j}, a_{i,j}^{\pi_i}) \sqrt{\frac{\mathbb{I}(y \in [y]_{i,j})}{N_i(x_{i,j}, a_{i,j}^{\pi_i}) P(y|x_{i,j}, a_{i,j}^{\pi_i})}} \widetilde{\Delta}_{i,j+1}(y) - \sqrt{\frac{\mathbb{I}(y \in [y]_{i,j})}{N_i(x_{i,j}, a_{i,j}^{\pi_i}) P(y|x_{i,j}, a_{i,j}^{\pi_i})}} \widetilde{\Delta}_{i,j+1}(x_{i,j+1}),$$

where:

$$[y]_{k,h} := \{y \in \mathcal{S} : N_k(x_{k,h}, a_{k,h}^{\pi_k}) P(y|x_{k,h}, a_{k,h}^{\pi_k}) \geqslant 2H^2 L\}.$$

Under event $\mathcal{E}$ the following events hold:

$$\mathcal{E}_{\mathrm{az}}(\mathcal{F}_{\widetilde{\Delta}, k, h}, H, L), \quad \text{and} \quad \mathcal{E}_{\mathrm{az}}(\mathcal{F}'_{\widetilde{\Delta}, k, h}, 1/\sqrt{L}, L).$$

Event $\mathcal{E}_{\mathrm{az}}(\mathcal{F}_{\widetilde{\Delta}, k, h}, H, L)$ is defined as the event such that:

$$\sum_{i=1}^{k} \sum_{j=h}^{H-1} \left[ \sum_{y \in \mathcal{S}} P(y|x_{i,j}, a_{i,j}^{\pi_i}) \widetilde{\Delta}_{i,j+1}(y) - \widetilde{\Delta}_{i,j+1}(x_{i,j+1}) \right] \leqslant 2\sqrt{k(H-1-h)H^2 L}$$

$$\leqslant 2\sqrt{H^2 T_k L}.$$

Under this event, we can apply the definition of $\varepsilon_{i,j}$ and derive that:

$$\sum_{i=1}^{k} \sum_{j=h}^{H-1} \varepsilon_{i,j} \leqslant 2\sqrt{H^2 T_k L}.$$

Event $\mathcal{E}_{\mathrm{az}}(\mathcal{F}'_{\widetilde{\Delta}, k, h}, 1/\sqrt{L}, L)$, on the other hand, is defined as the event such that:

$$\sum_{i=1}^{k} \sum_{j=h}^{H} \left[ \sum_{y \in \mathcal{S}} P(y|x_{i,j}, a_{i,j}^{\pi_i}) \sqrt{\frac{\mathbb{I}(y \in [y]_{i,j})}{N_i(x_{i,j}, a_{i,j}^{\pi_i}) P(y|x_{i,j}, a_{i,j}^{\pi_i})}} \widetilde{\Delta}_{i,j+1}(y) \right]$$

$$- \sqrt{\frac{\mathbb{I}(y \in [y]_{i,j})}{N_i(x_{i,j}, a_{i,j}^{\pi_i}) P(y|x_{i,j}, a_{i,j}^{\pi_i})}} \widetilde{\Delta}_{i,j+1}(x_{i,j+1})$$

$$\leqslant 2\sqrt{k(H-h) \frac{1}{\sqrt{L}^2} L}$$

$$\leqslant 2\sqrt{T_k}.$$

Under this event, we can apply the definition of $\overline{\varepsilon}_{i,j}$ and derive that:

$$\sum_{i=1}^{k} \sum_{j=h}^{H-1} \overline{\varepsilon}_{i,j} \leqslant 2\sqrt{T_k},$$

thus concluding the proof. $\qquad\square$

**Lemma D.4.** *Let $k \in [\![K]\!]$, $h \in [\![H]\!]$, and $x \in \mathcal{S}$. Let events $\mathcal{E}$ and $\Omega_{k,h}$ hold. Then the following bounds hold:*

$$\sum_{i=1}^{k} \mathbb{I}(x_{i,h} = x) \sum_{j=h}^{H} \varepsilon_{i,j} \leqslant 2\sqrt{H^2 T_k L},$$

$$\sum_{i=1}^{k} \mathbb{I}(x_{i,h} = x) \sum_{j=h}^{H} \overline{\varepsilon}_{i,j} \leqslant 2\sqrt{T_k},$$

*where $T_k = kH$.*

*Proof.* In a similar way to the proof of Lemma D.3, we recall the definitions of $\varepsilon_{i,j}$ and $\overline{\varepsilon}_{i,j}$:

$$\varepsilon_{i,j} := P(\cdot|x_{i,j}, a_{i,j}^{\pi_i})^{\mathsf{T}} \widetilde{\Delta}_{i,j+1}(\cdot) - \widetilde{\Delta}_{i,j+1}(x_{i,j+1}),$$

$$\overline{\varepsilon}_{i,j} := \sum_{y \in \mathcal{S}} P(y|x_{i,j}, a_{i,j}^{\pi_i}) \sqrt{\frac{\mathbb{I}(y \in [y]_{i,j})}{N_i(x_{i,j}, a_{i,j}^{\pi_i}) P(y|x_{i,j}, a_{i,j}^{\pi_i})}} \widetilde{\Delta}_{i,j+1}(y)$$

$$- \sqrt{\frac{\mathbb{I}(y \in [y]_{i,j})}{N_i(x_{i,j}, a_{i,j}^{\pi_i}) P(y|x_{i,j}, a_{i,j}^{\pi_i})}} \widetilde{\Delta}_{i,j+1}(x_{i,j+1}),$$

where:

$$[y]_{k,h} := \{y \in \mathcal{S} : N_k(x_{k,h}, a_{k,h}^{\pi_k}) P(y|x_{k,h}, a_{k,h}^{\pi_k}) \geqslant 2H^2 L\}.$$

Under event $\mathcal{E}$ the following events hold:

$$\mathcal{E}_{\mathrm{az}}(\mathcal{F}_{\widetilde{\Delta}, k, h, x}, H, L), \quad \text{and} \quad \mathcal{E}_{\mathrm{az}}(\mathcal{F}'_{\widetilde{\Delta}, k, h, x}, 1/\sqrt{L}, L).$$

Event $\mathcal{E}_{\mathrm{az}}(\mathcal{F}_{\widetilde{\Delta}, k, h, x}, H, L)$ is defined as the event such that:

$$\sum_{i=1}^{k} \mathbb{I}(x_{i,h} = x) \sum_{j=h}^{H-1} \left[ \sum_{y \in \mathcal{S}} P(y|x_{i,j}, a_{i,j}^{\pi_i}) \widetilde{\Delta}_{i,j+1}(y) - \widetilde{\Delta}_{i,j+1}(x_{i,j+1}) \right] \leqslant 2\sqrt{H^3 N'_{k,h}(x) L}.$$

Under this event, we can apply the definition of $\varepsilon_{i,j}$ and derive that:

$$\sum_{i=1}^{k} \mathbb{I}(x_{i,h} = x) \sum_{j=h}^{H-1} \varepsilon_{i,j} \leqslant 2\sqrt{H^3 N'_{k,h}(x) L}.$$

Event $\mathcal{E}_{\mathrm{az}}(\mathcal{F}'_{\widetilde{\Delta}, k, h, x}, 1/\sqrt{L}, L)$, on the other hand, is defined as the event such that:

$$\sum_{i=1}^{k} \mathbb{I}(x_{i,h} = x) \sum_{j=h}^{H} \left[ \sum_{y \in \mathcal{S}} P(y|x_{i,j}, a_{i,j}^{\pi_i}) \sqrt{\frac{\mathbb{I}(y \in [y]_{i,j})}{N_i(x_{i,j}, a_{i,j}^{\pi_i}) P(y|x_{i,j}, a_{i,j}^{\pi_i})}} \widetilde{\Delta}_{i,j+1}(y) \right]$$

$$- \sqrt{\frac{\mathbb{I}(y \in [y]_{i,j})}{N_i(x_{i,j}, a_{i,j}^{\pi_i}) P(y|x_{i,j}, a_{i,j}^{\pi_i})}} \widetilde{\Delta}_{i,j+1}(x_{i,j+1})$$

$$\leqslant 2\sqrt{N'_{k,h}(x)(H-h) \frac{1}{\sqrt{L}^2} L}$$

$$\leqslant 2\sqrt{H N'_{k,h}(x)}.$$

Under this event, we can apply the definition of $\overline{\varepsilon}_{i,j}$ and derive that:

$$\sum_{i=1}^{k} \mathbb{I}(x_{i,h} = x) \sum_{j=h}^{H-1} \overline{\varepsilon}_{i,j} \leqslant 2\sqrt{HN'_{k,h}(x)},$$

thus concluding the proof.

$\square$

**Lemma D.5.** *Let $k \in [\![K]\!]$ and $h \in [\![H]\!]$. Let $\pi_k$ be the policy followed during episode $k$. Under the events $\mathcal{E}$ and $\Omega_{k,h}$, the following holds for every $x \in \mathcal{S}$:*

$$\sum_{i=1}^{k} \sum_{j=h}^{H-1} \mathbb{V}_{j+1}^{\pi_k}(x_{i,j}, a_{i,j}^{\pi_i}) \leqslant HT_k + 2\sqrt{H^4 T_k L} + \frac{4}{3}H^3 L,$$

$$\sum_{i=1}^{k} \mathbb{I}(x_{i,h} = x) \sum_{j=h}^{H-1} \mathbb{V}_{j+1}^{\pi_k}(x_{i,j}, a_{i,j}^{\pi_i}) \leqslant H^2 N'_{k,h}(x) + 2\sqrt{H^5 N'_{k,h}(x)L} + \frac{4}{3}H^3 L.$$

*Proof.* We begin the proof by restating the definition of $\mathbb{V}_{j+1}^{\pi_k}(x_{i,j}, a_{i,j}^{\pi_i})$:

$$\mathbb{V}_{j+1}^{\pi_k}(x_{i,j}, a_{i,j}^{\pi_i}) := \operatorname*{\mathbb{V}ar}_{y \sim P(\cdot | x_{i,j}, a_{i,j}^{\pi_i})} [V_{j+1}^{\pi_k}(y)]$$

Under event $\mathcal{E}$, the following events hold:

$$\mathcal{E}_{\mathrm{fr}}(\mathcal{G}_{-\mathbb{V},k,h}, H^4 T_k, H^3, L) \quad \text{and} \quad \mathcal{E}_{\mathrm{fr}}(\mathcal{G}_{-\mathbb{V},k,h,x}, H^5 N'_{k,h}, H^3, L).$$

Event $\mathcal{E}_{\mathrm{fr}}(\mathcal{G}_{-\mathbb{V},k,h}, H^4 T_k, H^3, L)$ is defined as the event such that:

$$\sum_{i=1}^{k} \sum_{j=h}^{H-1} \mathbb{V}_{j+1}^{\pi_k}(x_{i,j}, a_{i,j}^{\pi_i}) - \sum_{i=1}^{k} \mathbb{E}\left[ \sum_{j=h}^{H-1} \mathbb{V}_{j+1}^{\pi_k}(x_{i,j}, a_{i,j}^{\pi_i}) | \mathcal{H}_{k,h} \right] \leqslant 2\sqrt{H^4 T_k L} + \frac{4H^3 L}{3},$$

which implies that:

$$\sum_{i=1}^{k} \sum_{j=h}^{H-1} \mathbb{V}_{j+1}^{\pi_k}(x_{i,j}, a_{i,j}^{\pi_i}) \leqslant \sum_{i=1}^{k} \mathbb{E}\left[ \sum_{j=h}^{H-1} \mathbb{V}_{j+1}^{\pi_k}(x_{i,j}, a_{i,j}^{\pi_i}) | \mathcal{H}_{k,h} \right] + 2\sqrt{H^4 T_k L} + \frac{4H^3 L}{3}. \tag{15}$$

On the other hand, event $\mathcal{E}_{\mathrm{fr}}(\mathcal{G}_{-\mathbb{V},k,h,x}, H^5 N'_{k,h}, H^3, L)$ is defined as the event such that:

$$\sum_{i=1}^{k} \mathbb{I}(x_{i,h} = x) \sum_{j=h}^{H-1} \mathbb{V}_{j+1}^{\pi_k}(x_{i,j}, a_{i,j}^{\pi_i}) - \sum_{i=1}^{k} \mathbb{I}(x_{i,h} = x) \mathbb{E}\left[ \sum_{j=h}^{H-1} \mathbb{V}_{j+1}^{\pi_k}(x_{i,j}, a_{i,j}^{\pi_i}) | \mathcal{H}_{k,h} \right]$$

$$\leqslant 2\sqrt{H^5 N'_{k,h}(x)L} + \frac{4H^3 L}{3},$$

which implies that:

$$\sum_{i=1}^{k} \mathbb{I}(x_{i,h} = x) \sum_{j=h}^{H-1} \mathbb{V}_{j+1}^{\pi_k}(x_{i,j}, a_{i,j}^{\pi_i}) \leqslant \sum_{i=1}^{k} \mathbb{I}(x_{i,h} = x) \mathbb{E}\left[\sum_{j=h}^{H-1} \mathbb{V}_{j+1}^{\pi_k}(x_{i,j}, a_{i,j}^{\pi_i}) \Big| \mathcal{H}_{k,h}\right] + 2\sqrt{H^5 N_{k,h}'(x) L} + \frac{4H^3 L}{3}.$$

$$(16)$$

Observe that by applying the *Law of Total Variance* (LTV) (see e.g., Theorem 9.5.4 of Blitzstein & Hwang, 2019), we can write:

$$\operatorname*{Var}_{x_{i,h+1},\ldots,x_{i,H-1}}\left[\sum_{j=h}^{H-1} R^\pi(x_{i,j})\right] = \underbrace{\operatorname*{Var}_{x_{i,h+1}}\left[\operatorname*{\mathbb{E}}_{x_{i,h+2},\ldots,x_{i,H-1}}\left[\sum_{j=h}^{H-1} R^\pi(x_{i,j})\Big|x_{i,h+1}\right]\right]}_{(a)}$$

$$(17)$$

$$+ \operatorname*{\mathbb{E}}_{x_{i,h+1}}\left[\underbrace{\operatorname*{Var}_{x_{i,h+2},\ldots,x_{i,H-1}}\left[\sum_{j=h}^{H-1} R^\pi(x_{i,j})\Big|x_{i,h+1}\right]}_{(b)}\right].$$

Term $(a)$ can be rewritten as:

$$(a) = \operatorname*{Var}_{x_{i,h+1}}\left[R^\pi(x_{i,h}) + \operatorname*{\mathbb{E}}_{x_{i,h+2},\ldots,x_{i,H-1}}\left[\sum_{j=h+1}^{H-1} R^\pi(x_{i,j})\Big|x_{i,h+1}\right]\right]$$

$$= \operatorname*{Var}_{x_{i,h+1}}\left[V_{h+1}^{\pi_k}(x_{i,h+1})\right] \qquad (18)$$

$$= \mathbb{V}_{h+1}^{\pi_k}(x_{i,h}, a_{i,h}^{\pi_i}), \qquad (19)$$

where Equation (18) is obtained by observing that $R^\pi(x_{i,h})$ has zero variance w.r.t. $x_{i,h+1}$, and by applying the definition of value function.

We can then recursively apply the LTV to term $(b)$ and, considering the expectation over the trajectory generated following policy $\pi$ from stage $h$ onward, we can write:

$$\operatorname*{Var}_{x_{i,h+1},\ldots,x_{i,H-1}}\left[\sum_{j=h}^{H-1} R^\pi(x_{i,j})\right] = \mathbb{E}\left[\sum_{j=h}^{H-1} \mathbb{V}_{j+1}^{\pi_k}(x_{i,j}, a_{i,j}^{\pi_i})\right]. \qquad (20)$$

By applying the result of Equation (20) to Equations (15) and (16), we get:

$$\sum_{i=1}^{k} \mathbb{E}\left[\sum_{j=h}^{H-1} \mathbb{V}_{j+1}^{\pi_k}(x_{i,j}, a_{i,j}^{\pi_i}) \Big| \mathcal{H}_{k,h}\right] = \sum_{i=1}^{k} \operatorname*{Var}\left[\sum_{j=h+1}^{H-1} R^\pi(x_{i,j})\right]$$

$$\leqslant k(H-h)^2$$

$$\leqslant HT_k, \qquad (21)$$

and:

$$\sum_{i=1}^{k} \mathbb{I}(x_{i,h} = x)\mathbb{E}\left[\sum_{j=h}^{H-1} \mathbb{V}_{j+1}^{\pi_k}(x_{i,j}, a_{i,j}^{\pi_i})|\mathcal{H}_{k,h}\right] = \sum_{i=1}^{k} \mathbb{I}(x_{i,h} = x)\,\mathbb{V}\mathrm{ar}\left[\sum_{j=h+1}^{H-1} R^{\pi}(x_{i,j})\right]$$
$$\leqslant N_{k,h}'(x)(H-h)^2$$
$$\leqslant H^2 N_{k,h}'(x). \tag{22}$$

Finally, we can plug Equations (21) and (22) into Equations (15) and (16), respectively, obtaining:

$$\sum_{i=1}^{k}\sum_{j=h}^{H-1} \mathbb{V}_{j+1}^{\pi_k}(x_{i,j}, a_{i,j}^{\pi_i}) \leqslant HT_k + 2\sqrt{H^4 T_k L} + \frac{4}{3}H^3 L,$$

$$\sum_{i=1}^{k} \mathbb{I}(x_{i,h} = x)\sum_{j=h}^{H-1} \mathbb{V}_{j+1}^{\pi_k}(x_{i,j}, a_{i,j}^{\pi_i}) \leqslant H^2 N_{k,h}'(x) + 2\sqrt{H^5 N_{k,h}'(x)L} + \frac{4}{3}H^3 L,$$

thus concluding the proof. $\qquad\square$

**Lemma D.6.** *Let $k \in [\![K]\!]$ and $h \in [\![H]\!]$. Let $\pi_k$ be the policy played during episode $k$. Under the events $\mathcal{E}$ and $\Omega_{k,h}$, the following holds for every $x \in \mathcal{S}$:*

$$\sum_{i=1}^{k}\sum_{j=h}^{H-1}\left(\mathbb{V}_{j+1}^{*}(x_{i,j}, a_{i,j}^{\pi_i}) - \mathbb{V}_{j+1}^{\pi_k}(x_{i,j}, a_{i,j}^{\pi_i})\right) \leqslant 2H\sum_{i=1}^{k}\sum_{j=h}^{H-1} \widetilde{\Delta}_{i,j}(x_{i,j}) + 4H^2\sqrt{T_k L},$$

$$\sum_{i=1}^{k} \mathbb{I}(x_{i,h} = x)\sum_{j=h}^{H-1}\left(\mathbb{V}_{j+1}^{*}(x_{i,j}, a_{i,j}^{\pi_i}) - \mathbb{V}_{j+1}^{\pi_k}(x_{i,j}, a_{i,j}^{\pi_i})\right) \leqslant 2H\sum_{i=1}^{k} \mathbb{I}(x_{i,h} = x)\sum_{j=h}^{H-1} \widetilde{\Delta}_{i,h}(x_{i,h}) + 4H^2\sqrt{HN_{k,h}'(x)L}.$$

*Proof.* We demonstrate the result by providing an upper bound to $\mathbb{V}_{j+1}^{*} - \mathbb{V}_{j+1}^{\pi_k}$ first, and then bounding its summation over episodes and stages. We can demonstrate that:

$$\mathbb{V}_{j+1}^{*}(x_{i,j}, a_{i,j}^{\pi_i}) - \mathbb{V}_{j+1}^{\pi_k}(x_{i,j}, a_{i,j}^{\pi_i}) = \mathbb{V}\mathrm{ar}_{y \sim P(\cdot|x_{i,j}, a_{i,j}^{\pi_i})}[V_{j+1}^{*}(y)] - \mathbb{V}\mathrm{ar}_{y \sim P(\cdot|x_{i,j}, a_{i,j}^{\pi_i})}[V_{j+1}^{\pi}(y)]$$
$$\leqslant \mathbb{E}_{y \sim P(\cdot|x_{i,j}, a_{i,j}^{\pi_i})}[(V_{j+1}^{*}(y))^2 - (V_{j+1}^{\pi}(y))^2] \tag{23}$$
$$\leqslant 2H\mathbb{E}_{y \sim P(\cdot|x_{i,j}, a_{i,j}^{\pi_i})}[V_{j+1}^{*}(y) - V_{j+1}^{\pi}(y)], \tag{24}$$

where Equation (23) is obtained by applying the definition of variance and observing that $V_{j+1}^{*}(x) \geqslant V_{j+1}^{\pi}(x)$ by definition, and Equation (24) is obtained by expanding the square and by observing that $V_{j+1}^{\pi}(x) \leqslant V_{j+1}^{*}(x) \leqslant H$.

Using the argument of Equation (24), we obtain the following inequalities:

$$\sum_{i=1}^{k}\sum_{j=h}^{H-1}\left(\mathbb{V}_{j+1}^{*}(x_{i,j}, a_{i,j}^{\pi_i}) - \mathbb{V}_{j+1}^{\pi_k}(x_{i,j}, a_{i,j}^{\pi_i})\right) \leqslant 2H\underbrace{\sum_{i=1}^{k}\sum_{j=h}^{H-1}\mathbb{E}_{y \sim P(\cdot|x_{i,j}, a_{i,j}^{\pi_i})}[\Delta_{i,j+1}(y)]}_{(a)}, \tag{25}$$

$$\sum_{i=1}^{k} \mathbb{I}(x_{i,h} = x)\sum_{j=h}^{H-1}\left(\mathbb{V}_{j+1}^{*}(x_{i,j}, a_{i,j}^{\pi_i}) - \mathbb{V}_{j+1}^{\pi_k}(x_{i,j}, a_{i,j}^{\pi_i})\right)$$
$$\leqslant 2H\underbrace{\sum_{i=1}^{k} \mathbb{I}(x_{i,h} = x)\sum_{j=h}^{H-1}\mathbb{E}_{y \sim P(\cdot|x_{i,j}, a_{i,j}^{\pi_i})}[\Delta_{i,j+1}(y)]}_{(b)}. \tag{26}$$

We now bound term $(a)$ as follows:

$$(a) \leqslant \sum_{i=1}^{k} \sum_{j=h}^{H-1} \mathbb{E}_{y \sim P(\cdot | x_{i,j}, a_{i,j}^{\pi_i})} [\widetilde{\Delta}_{i,j+1}(y)] \tag{27}$$

$$\leqslant 2\sqrt{H^2 T_k L} + \sum_{i=1}^{k} \sum_{j=h}^{H-1} \widetilde{\Delta}_{i,j+1}(x_{i,j+1}) \tag{28}$$

where Equation (27) is obtained because, under $\Omega_{k,h}$, it holds that $V_{j+1}^*(y) \leqslant V_{i,j+1}(y)$. Equation (28) is obtained by considering that, under event $\mathcal{E}$, the event $\mathcal{E}_{\mathrm{az}}(\mathcal{F}_{\widetilde{\Delta},k,h}, H, L)$ holds, as shown in Lemma D.3.

Following a similar procedure, we bound term $(b)$ by considering event $\mathcal{E}_{\mathrm{az}}(\mathcal{F}_{\widetilde{\Delta},k,h,x}, H, L)$, obtaining:

$$(b) \leqslant 2H\sqrt{HN'_{k,h}(x)L} + \sum_{i=1}^{k} \mathbb{I}(x_{i,h} = x) \sum_{j=h}^{H-1} \widetilde{\Delta}_{i,j+1}(x_{i,j+1}). \tag{29}$$

We can then plug Equations (28) and (29) into Equations (25) and (26), respectively, to write:

$$\sum_{i=1}^{k} \sum_{j=h}^{H-1} \left( \mathbb{V}_{j+1}^*(x_{i,j}, a_{i,j}^{\pi_i}) - \mathbb{V}_{j+1}^{\pi_k}(x_{i,j}, a_{i,j}^{\pi_i}) \right) \leqslant 2H \sum_{i=1}^{k} \sum_{j=h}^{H-1} \widetilde{\Delta}_{i,j}(x_{i,j}) + 4H^2\sqrt{T_k L},$$

$$\sum_{i=1}^{k} \mathbb{I}(x_{i,h} = x) \sum_{j=h}^{H-1} \left( \mathbb{V}_{j+1}^*(x_{i,j}, a_{i,j}^{\pi_i}) - \mathbb{V}_{j+1}^{\pi_k}(x_{i,j}, a_{i,j}^{\pi_i}) \right) \leqslant 2H \sum_{i=1}^{k} \mathbb{I}(x_{i,h} = x) \sum_{j=h}^{H-1} \widetilde{\Delta}_{i,h}(x_{i,h}) + 4H^2\sqrt{HN'_{k,h}(x)L},$$

thus concluding the proof. $\qquad \square$

**Lemma D.7.** *Let $k \in [\![K]\!]$ and $h \in [\![H]\!]$. Let $\pi_k$ denote the policy followed during episode $k$. Under events $\mathcal{E}$ and $\Omega_{k,h}$, the following inequalities hold for every $x \in \mathcal{S}$:*

$$\sum_{i=1}^{k} \sum_{j=h}^{H-1} \widehat{\mathbb{V}}_{i,j+1}(x_{i,j}, a_{i,j}^{\pi_i}) - \mathbb{V}_{j+1}^{\pi_i}(x_{i,j}, a_{i,j}^{\pi_i}) \leqslant 7H^2 S\sqrt{AT_k L} + 2H \sum_{i=1}^{k} \sum_{j=h}^{H-1} \widetilde{\Delta}_{i,j+1}(x_{i,j+1}),$$

$$\sum_{i=1}^{k} \mathbb{I}(x_{i,j} = x) \sum_{j=h}^{H-1} \widehat{\mathbb{V}}_{i,j+1}(x_{i,j}, a_{i,j}^{\pi_i}) - \mathbb{V}_{j+1}^{\pi_i}(x_{i,j}, a_{i,j}^{\pi_i})$$

$$\leqslant 7H^2 S\sqrt{HAN'_{k,h}(x)L} + 2H \sum_{i=1}^{k} \mathbb{I}(x_{i,h} = x) \sum_{j=h}^{H-1} \widetilde{\Delta}_{i,j+1}(x_{i,j+1}).$$

*Proof.* Similarly to the proof of Lemma D.6, we demonstrate the result by providing an upper bound to $\widehat{\mathbb{V}}_{i,j+1} - \mathbb{V}_{j+1}^{\pi_k}$ first, and then bounding its summation over episodes and stages.

$$\widehat{\mathbb{V}}_{i,j+1}(x_{i,j}, a_{i,j}^{\pi_i}) - \mathbb{V}_{j+1}^{\pi_i}(x_{i,j}, a_{i,j}^{\pi_i})$$

$$= \underset{y \sim \widehat{P}_i(\cdot|x_{i,j}, a_{i,j}^{\pi_i})}{\mathbb{V}\text{ar}} [V_{i,j+1}(y)] - \underset{y \sim P(\cdot|x_{i,j}, a_{i,j}^{\pi_i})}{\mathbb{V}\text{ar}} [V_{j+1}^{\pi_i}(y)]$$

$$= \mathbb{E}_{y \sim \widehat{P}_i(\cdot|x_{i,j}, a_{i,j}^{\pi_i})}[(V_{i,j+1}(y))^2] - \mathbb{E}_{y \sim \widehat{P}_i(\cdot|x_{i,j}, a_{i,j}^{\pi_i})}[V_{i,j+1}(y)]^2$$

$$- \mathbb{E}_{y \sim P(\cdot|x_{i,j}, a_{i,j}^{\pi_i})}[(V_{j+1}^{\pi_i}(y))^2] + \mathbb{E}_{y \sim P(\cdot|x_{i,j}, a_{i,j}^{\pi_i})}[V_{j+1}^{\pi_i}(y)]^2$$

$$\leqslant \mathbb{E}_{y \sim \widehat{P}_i(\cdot|x_{i,j}, a_{i,j}^{\pi_i})}[(V_{i,j+1}(y))^2] - \mathbb{E}_{y \sim P(\cdot|x_{i,j}, a_{i,j}^{\pi_i})}[V_{j+1}^{\pi_i}(y)]^2$$

$$+ \mathbb{E}_{y \sim P(\cdot|x_{i,j}, a_{i,j}^{\pi_i})}[V_{j+1}^*(y)]^2 - \mathbb{E}_{y \sim \widehat{P}_i(\cdot|x_{i,j}, a_{i,j}^{\pi_i})}[V_{j+1}^*(y)]^2 \tag{30}$$

$$\leqslant \mathbb{E}_{y \sim \widehat{P}_i(\cdot|x_{i,j}, a_{i,j}^{\pi_i})}[(V_{i,j+1}(y))^2] - \mathbb{E}_{y \sim P(\cdot|x_{i,j}, a_{i,j}^{\pi_i})}[(V_{i,j+1}(y))^2]$$

$$+ \mathbb{E}_{y \sim P(\cdot|x_{i,j}, a_{i,j}^{\pi_i})}[(V_{i,j+1}(y))^2] - \mathbb{E}_{y \sim P(\cdot|x_{i,j}, a_{i,j}^{\pi_i})}[V_{j+1}^{\pi_i}(y)]^2$$

$$+ 2H \sum_{y \in \mathcal{S}} (P(y|x_{i,j}, a_{i,j}^{\pi_i}) - \widehat{P}_i(y|x_{i,j}, a_{i,j}^{\pi_i})) V_{j+1}^*(y) \tag{31}$$

$$\leqslant \mathbb{E}_{y \sim \widehat{P}_i(\cdot|x_{i,j}, a_{i,j}^{\pi_i})}[(V_{i,j+1}(y))^2] - \mathbb{E}_{y \sim P(\cdot|x_{i,j}, a_{i,j}^{\pi_i})}[(V_{i,j+1}(y))^2]$$

$$+ \mathbb{E}_{y \sim P(\cdot|x_{i,j}, a_{i,j}^{\pi_i})}[(V_{i,j+1}(y))^2] - \mathbb{E}_{y \sim P(\cdot|x_{i,j}, a_{i,j}^{\pi_i})}[V_{j+1}^{\pi_i}(y)]^2 + 4H\sqrt{\frac{H^2 L}{N_i(x_{i,j}, a_{i,j}^{\pi_i})}} \tag{32}$$

where Equation (30) follows from the fact that, under $\Omega_{k,h}$, $V_{i,j}(y) \geqslant V_j^*(y) \geqslant V_j^{\pi_i}(y)$. Equation (31) is obtained by adding and subtracting $\mathbb{E}_{y \sim P(\cdot|x_{i,j}, a_{i,j}^{\pi_i})}[(V_{i,j+1}(y))^2]$, and by observing that $V_j^*(y) \leqslant H$. Equation (32) is obtained by bounding the model error via Hoeffding's inequality.

Putting this result into the double summation, we get:

$$\sum_{i=1}^{k} \sum_{j=h}^{H-1} \widehat{\mathbb{V}}_{i,j+1}(x_{i,j}, a_{i,j}^{\pi_i}) - \mathbb{V}_{j+1}^{\pi_i}(x_{i,j}, a_{i,j}^{\pi_i})$$

$$\leqslant \underbrace{\sum_{i=1}^{k} \sum_{j=h}^{H-1} \left[ \mathbb{E}_{y \sim \widehat{P}_i(\cdot|x_{i,j}, a_{i,j}^{\pi_i})}[(V_{i,j+1}(y))^2] - \mathbb{E}_{y \sim P(\cdot|x_{i,j}, a_{i,j}^{\pi_i})}[(V_{i,j+1}(y))^2] \right]}_{(a)} \tag{33}$$

$$+ \underbrace{\sum_{i=1}^{k} \sum_{j=h}^{H-1} \mathbb{E}_{y \sim P(\cdot|x_{i,j}, a_{i,j}^{\pi_i})}[(V_{i,j+1}(y))^2] - \mathbb{E}_{y \sim P(\cdot|x_{i,j}, a_{i,j}^{\pi_i})}[V_{j+1}^{\pi_i}(y)]^2}_{(b)} + \underbrace{\sum_{i=1}^{k} \sum_{j=h}^{H-1} 4H\sqrt{\frac{H^2 L}{N_i(x_{i,j}, a_{i,j}^{\pi_i})}}}_{(c)}.$$

We begin by bounding term $(a)$:

$$
\begin{aligned}
(a) &\leqslant \sum_{i=1}^{k} \sum_{j=h}^{H-1} H^2 \|\widehat{P}_i(\cdot|x_{i,j}, a_{i,j}^{\pi_i}) - P(\cdot|x_{i,j}, a_{i,j}^{\pi_i})\|_1 \\
&\leqslant \sum_{i=1}^{k} \sum_{j=h}^{H-1} 2H^2 \sqrt{\frac{SL}{N_i(x_{i,j}, a_{i,j}^{\pi_i})}} \\
&= 2H^2\sqrt{SL} \sum_{x \in \mathcal{S}} \sum_{a \in \mathcal{A}} \sum_{n=1}^{N_i(x,a)} n^{-1/2} \\
&\leqslant 2H^2\sqrt{SL} \sum_{x \in \mathcal{S}} \sum_{a \in \mathcal{A}} \sum_{n=1}^{\frac{kH}{SA}} n^{-1/2} \\
&\leqslant H^2 S \sqrt{AT_kL},
\end{aligned}
\tag{34}
$$

where Equation (34) follows by applying the result of Theorem 2.1 of (Weissman et al., 2003), which holds under event $\mathcal{E}$. We now bound term $(b)$:

$$
\begin{aligned}
(b) &= \sum_{i=1}^{k} \sum_{j=h}^{H-1} \mathbb{E}_{y \sim P(\cdot|x_{i,j}, a_{i,j}^{\pi_i})} [(V_{i,j+1}(y) + V_{j+1}^{\pi_k}(y))(V_{i,j+1}(y) - V_{j+1}^{\pi_k}(y))] \\
&\leqslant 2H \sum_{i=1}^{k} \sum_{j=h}^{H-1} \mathbb{E}_{y \sim P(\cdot|x_{i,j}, a_{i,j}^{\pi_i})} [\widetilde{\Delta}_{i,j+1}(y)] \\
&\leqslant 2H\Big(\sum_{i=1}^{k} \sum_{j=h}^{H-1} \widetilde{\Delta}_{i,j+1}(x_{i,j+1}) + 2H\sqrt{T_kL}\Big),
\end{aligned}
\tag{35}
$$

where Equation (35) is obtained because under events $\mathcal{E}$, also event $\mathcal{E}_{\mathrm{az}}(\mathcal{F}_{\widetilde{\Delta},k,h}, H, L)$ holds. We now bound term $(c)$:

$$
\begin{aligned}
(c) &\leqslant 4H^2\sqrt{L} \sum_{x \in \mathcal{S}} \sum_{a \in \mathcal{A}} \sum_{n=1}^{\frac{kH}{SA}} n^{-1/2} \\
&\leqslant 2H^2\sqrt{SAT_kL}.
\end{aligned}
$$

Finally, by plugging the bounds of terms $(a)$, $(b)$, and $(c)$ into Equation (33), we get:

$$
\begin{aligned}
\sum_{i=1}^{k} \sum_{j=h}^{H-1} &\widehat{\mathbb{V}}_{i,j+1}(x_{i,j}, a_{i,j}^{\pi_i}) - \mathbb{V}_{j+1}^{\pi_i}(x_{i,j}, a_{i,j}^{\pi_i}) \\
&\leqslant H^2 S\sqrt{AT_kL} + 2H \sum_{i=1}^{k} \sum_{j=h}^{H-1} \widetilde{\Delta}_{i,j+1}(x_{i,j+1}) \\
&\quad + 4H^2\sqrt{T_kL} + 2H^2\sqrt{SAT_kL} \\
&\leqslant 7H^2 S\sqrt{AT_kL} + 2H \sum_{i=1}^{k} \sum_{j=h}^{H-1} \widetilde{\Delta}_{i,j+1}(x_{i,j+1}).
\end{aligned}
$$

Using the same procedure, we can bound the following summation as:

$$\sum_{i=1}^{k} \mathbb{I}(x_{i,j} = x) \sum_{j=h}^{H-1} \widehat{\mathbb{V}}_{i,j+1}(x_{i,j}, a_{i,j}^{\pi_i}) - \mathbb{V}_{j+1}^{\pi_i}(x_{i,j}, a_{i,j}^{\pi_i})$$

$$\leqslant 7H^2 S\sqrt{HAN'_{k,h}(x)L} + 2H \sum_{i=1}^{k} \mathbb{I}(x_{i,h} = x) \sum_{j=h}^{H-1} \widetilde{\Delta}_{i,j+1}(x_{i,j+1}),$$

thus concluding the proof. $\qquad\square$

**Lemma D.8** (Summation over typical episodes of state-action wise model errors). *Let $k \in [\![K]\!]$ and $h \in [\![H]\!]$. Let $\pi_k$ be the policy followed during episode $k$. Under events $\mathcal{E}$ and $\Omega_{k,h}$, the following inequalities hold for every $x \in \mathcal{S}$:*

$$\sum_{i=1}^{k} \mathbb{I}(i \in [k]_{\text{typ}}) \sum_{j=h}^{H-1} \left[ \widehat{P}_i(\cdot|x_{i,j}, a_{i,j}^{\pi_i}) - P(\cdot|x_{i,j}, a_{i,j}^{\pi_i}) \right]^\top V_{j+1}^*(\cdot)$$

$$\leqslant \sqrt{6HSAT_k L^2} + \frac{2}{3}HSAL^2 \tag{36}$$

$$+ 2\sqrt{HSAL^2 (\sum_{i=1}^{k} \sum_{j=h}^{H-1} \widetilde{\Delta}_{i,j}(x_{i,j}))},$$

$$\sum_{i=1}^{k} \mathbb{I}(i \in [k]_{\text{typ},x}, x_{i,h} = x) \sum_{j=h}^{H-1} \left[ \widehat{P}_i(\cdot|x_{i,j}, a_{i,j}^{\pi_i}) - P(\cdot|x_{i,j}, a_{i,j}^{\pi_i}) \right]^\top V_{j+1}^*(\cdot)$$

$$\leqslant \sqrt{6H^2 SAN'_{k,h}(x)L^2} + \frac{2}{3}HSAL^2 \tag{37}$$

$$+ 2\sqrt{HSAL^2 (\sum_{i=1}^{k} \mathbb{I}(x_{i,h} = x) \sum_{j=h}^{H-1} \widetilde{\Delta}_{i,j}(x_{i,j}))},$$

*where:*

$$[k]_{\text{typ}} := \{i \in [\![k]\!] : (x_{i,h}, a_{i,h}^{\pi_i}) \in [(x,a)]_k, i \geqslant 250HS^2AL, \forall h \in [\![H]\!]\},$$

$$[k]_{\text{typ},x} := \{i \in [\![k]\!] : (x_{i,h}, a_{i,h}^{\pi_i}) \in [(x,a)]_k, N'_{k,h}(x) \geqslant 250HS^2AL, \forall h \in [\![H]\!]\},$$

$$[(x,a)]_k := \{(x,a) \in \mathcal{S} \times \mathcal{A} : N_k(x,a) \geqslant H, N'_{k,h}(x) \geqslant H, \forall h \in [\![H]\!]\}.$$

*Proof.* We begin by demonstrating the bound of Equation (36):

$$\sum_{i=1}^{k} \mathbb{I}(i \in [k]_{\text{typ}}) \sum_{j=h}^{H-1} \left[\widehat{P}_i(\cdot|x_{i,j}, a_{i,j}^{\pi_i}) - P(\cdot|x_{i,j}, a_{i,j}^{\pi_i})\right]^{\top} V_{j+1}^{*}(\cdot)$$

$$\leqslant \sum_{i=1}^{k} \mathbb{I}(i \in [k]_{\text{typ}}) \sum_{j=h}^{H-1} \left[\sqrt{\frac{2\mathbb{V}_{j+1}^{*}(x_{i,j}, a_{i,j}^{\pi_i})L}{N_i(x_{i,j}, a_{i,j}^{\pi_i})}} + \frac{2HL}{3N_i(x_{i,j}, a_{i,j}^{\pi_i})}\right] \tag{38}$$

$$\leqslant \sqrt{2L} \underbrace{\sqrt{\sum_{i=1}^{k} \sum_{j=h}^{H-1} \mathbb{V}_{j+1}^{*}(x_{i,j}, a_{i,j}^{\pi_i})}}_{(a)} \underbrace{\sqrt{\sum_{i=1}^{k} \mathbb{I}(i \in [k]_{\text{typ}}) \sum_{j=h}^{H-1} \frac{1}{N_i(x_{i,j}, a_{i,j}^{\pi_i})}}}_{(b)}$$

$$+ \underbrace{\sum_{i=1}^{k} \mathbb{I}(i \in [k]_{\text{typ}}) \sum_{j=h}^{H-1} \frac{2HL}{3N_i(x_{i,j}, a_{i,j}^{\pi_i})}}_{(c)}, \tag{39}$$

where Equation (38) is obtained by applying Bernstein's inequality (see, e.g., Cesa-Bianchi & Lugosi, 2006), and Equation (39) is obtained by apllying Cauchy-Schwarz's inequality. We now bound terms $(a)$, $(b)$, and $(c)$.

By adding and subtracting $\mathbb{V}_{j+1}^{\pi_i}(x_{i,j}, a_{i,j}^{\pi_i})$ to term $(a)$, we can rewrite it as:

$$(a) = \underbrace{\sum_{i=1}^{k} \sum_{j=h}^{H-1} \mathbb{V}_{j+1}^{\pi_i}(x_{i,j}, a_{i,j}^{\pi_i})}_{(d)} + \underbrace{\sum_{i=1}^{k} \sum_{j=h}^{H-1} \left(\mathbb{V}_{j+1}^{*}(x_{i,j}, a_{i,j}^{\pi_i}) - \mathbb{V}_{j+1}^{\pi_i}(x_{i,j}, a_{i,j}^{\pi_i})\right)}_{(e)}.$$

As events $\mathcal{E}$ and $\Omega_{k,h}$ hold, we can apply Lemmas D.5 and D.6 to bound terms $(d)$ and $(e)$, respectively, thus obtaining:

$$(a) \leqslant HT_k + 2H^2\sqrt{T_k L} + \frac{4}{3}H^3 L + 2H \sum_{i=1}^{k} \sum_{j=h}^{H-1} \widetilde{\Delta}_{i,j}(x_{i,j}) + 4H^2\sqrt{T_k L}$$

$$\leqslant 3T_k H + 2H \sum_{i=1}^{k} \sum_{j=h}^{H-1} \widetilde{\Delta}_{i,j}(x_{i,j}), \tag{40}$$

where Equation (40) holds under the condition of $[k]_{\text{typ}}$.

We now bound terms $(b)$ and $(c)$ as follows:

$$(b) \leqslant \sum_{x \in \mathcal{S}} \sum_{a \in \mathcal{A}} \sum_{n=1}^{kH} n^{-1}$$

$$\leqslant SAL, \tag{41}$$

$$(c) \leqslant \frac{2}{3}HL \sum_{x \in \mathcal{S}} \sum_{a \in \mathcal{A}} \sum_{n=1}^{kH} n^{-1}$$

$$\leqslant \frac{2}{3}HSAL^2. \tag{42}$$

Finally, by plugging the results of Equations (40), (41), and (42) into Equation (39), we get:

$$\sum_{i=1}^{k} \mathbb{I}(i \in [k]_{\text{typ}}) \sum_{j=h}^{H-1} \left[ \widehat{P}_i(\cdot | x_{i,j}, a_{i,j}^{\pi_i}) - P(\cdot | x_{i,j}, a_{i,j}^{\pi_i}) \right]^{\top} V_{j+1}^*(\cdot)$$

$$\leqslant \sqrt{2L} \sqrt{3 T_k H + 2H \sum_{i=1}^{k} \sum_{j=h}^{H-1} \widetilde{\Delta}_{i,j}(x_{i,j}) \sqrt{SAL} + \frac{2}{3} HSAL^2} \tag{43}$$

$$\leqslant \sqrt{6 HSAT_k L^2} + 2 \sqrt{HSAL^2 (\sum_{i=1}^{k} \sum_{j=h}^{H-1} \widetilde{\Delta}_{i,j}(x_{i,j})) + \frac{2}{3} HSAL^2}, \tag{44}$$

where Equation (44) is obtained by computing the product of the square roots and by the subadditivity of the square root. Following the same procedure, we can obtain the upper bound of Equation (37) by substituting terms $T_k$ with $HN'_{k,h}(x)$, and terms $\sum_{i=1}^{k} \sum_{j=h}^{H-1} \widetilde{\Delta}_{i,j}(x_{i,j})$ with $\sum_{i=1}^{k} \mathbb{I}(x_{i,h} = x) \sum_{j=h}^{H-1} \widetilde{\Delta}_{i,j}(x_{i,j})$. $\qquad\square$

**Lemma D.9** (Summation over typical episodes of bonus terms). *Let $k \in [\![K]\!]$ and $h \in [\![H]\!]$. Let $\pi_k$ be the policy followed during episode $k$. Let the UCB bonus be defined as:*

$$b_{k,h}(x,a) = \sqrt{\frac{4L \operatorname{Var}_{y \sim \widehat{P}_k(\cdot | x, a)}[V_{k,h+1}(y)]}{N_k(x,a)}} + \frac{7HL}{3(N_k(x,a) - 1)}$$

$$+ \sqrt{\frac{4 \min\left\{ \mathbb{E}_{y \sim \widehat{P}_k(\cdot | x, a)}\left[ \frac{84^2 H^3 S^2 AL^2}{N'_{k,h+1}(y)} \right], H^2 \right\}}{N_k(x,a)}}.$$

*Under the events $\mathcal{E}$ and $\Omega_{k,h}$ the following inequalities hold for every $x \in \mathcal{S}$:*

$$\sum_{i=1}^{k} \mathbb{I}(i \in [k]_{\text{typ}}) \sum_{j=h}^{H-1} b_{i,j}(x_{i,j}, a_{i,j}^{\pi_i})$$

$$\leqslant \sqrt{28 HSAT_k L^2} + \frac{7}{3} HSAL^2 + 2\sqrt{84^2 H^3 S^4 A^2 L^4} \tag{45}$$

$$+ \sqrt{8 HSAL^2 \sum_{i=1}^{k} \sum_{j=h}^{H-1} \widetilde{\Delta}_{i,j+1}(x_{i,j+1})},$$

$$\sum_{i=1}^{k} \mathbb{I}(i \in [k]_{\text{typ},x}, x_{i,h} = x) \sum_{j=h}^{H-1} b_{i,j}(x_{i,j}, a_{i,j}^{\pi_i})$$

$$\leqslant \sqrt{28 H^2 SAN'_{k,h}(x) L^2} + \frac{7}{3} HSAL^2 + 2\sqrt{84^2 H^3 S^4 A^2 L^4} \tag{46}$$

$$+ \sqrt{8 HSAL^2 \sum_{i=1}^{k} \mathbb{I}(x_{i,h} = x) \sum_{j=h}^{H-1} \widetilde{\Delta}_{i,j+1}(x_{i,j+1})},$$

*where:*

$$[k]_{\text{typ}} := \{i \in [\![k]\!] : (x_{i,h}, a_{i,h}^{\pi_i}) \in [(x,a)]_k, i \geqslant 250 HS^2 AL, \forall h \in [\![H]\!]\},$$

$$[k]_{\text{typ},x} := \{i \in [\![k]\!] : (x_{i,h}, a_{i,h}^{\pi_i}) \in [(x,a)]_k, N'_{k,h}(x) \geqslant 250 HS^2 AL, \forall h \in [\![H]\!]\},$$

$$[(x,a)]_k := \{(x,a) \in \mathcal{S} \times \mathcal{A} : N_k(x,a) \geqslant H, N'_{k,h}(x) \geqslant H, \forall h \in [\![H]\!]\}.$$

*Proof.* We begin by demonstrating the bound of Equation (45). We can rewrite the summation as:

$$\sum_{i=1}^{k} \mathbb{I}(i \in [k]_{\text{typ}}) \sum_{j=h}^{H-1} b_{i,j}(x_{i,j}, a_{i,j}^{\pi_i}) = \underbrace{\sum_{i=1}^{k} \mathbb{I}(i \in [k]_{\text{typ}}) \sum_{j=h}^{H-1} \sqrt{\frac{4L\widehat{\mathbb{V}}_{i,j+1}(x_{i,j}, a_{i,j}^{\pi_i})}{N_i(x_{i,j}, a_{i,j}^{\pi_i})}}}_{(a)}$$

$$+ \underbrace{\sum_{i=1}^{k} \mathbb{I}(i \in [k]_{\text{typ}}) \sum_{j=h}^{H-1} \frac{7HL}{3(N_i(x_{i,j}, a_{i,j}^{\pi_i}) - 1)}}_{(b)} \tag{47}$$

$$+ \underbrace{\sum_{i=1}^{k} \mathbb{I}(i \in [k]_{\text{typ}}) \sum_{j=h}^{H-1} \sqrt{\frac{4\mathbb{E}_{y \sim \widehat{P}_i(\cdot | x_{i,j}, a_{i,j}^{\pi_i})} b'_{i,j+1}(y)}{N_i(x_{i,j}, a_{i,j}^{\pi_i})}}}_{(c)},$$

where $b'_{i,j+1}(y) = \min\{\frac{84^2 H^2 S^2 AL^2}{N'_{i,j+1}(y)}, H^2\}$. First of all, we observe that we can bound term $(b)$ by using a pigeonhole argument as:

$$(b) \leqslant \frac{7}{3} HSAL^2. \tag{48}$$

We now bound term $(a)$. By applying Cauchy-Schwarz's inequality, we obtain:

$$(a) \leqslant \sqrt{4L} \sqrt{\underbrace{\sum_{i=1}^{k} \sum_{j=h}^{H-1} \widehat{\mathbb{V}}_{i,j+1}(x_{i,j}, a_{i,j}^{\pi_i})}_{(d)}} \sqrt{\underbrace{\sum_{i=1}^{k} \mathbb{I}(i \in [k]_{\text{typ}}) \sum_{j=h}^{H-1} \frac{1}{N_i(x_{i,j}, a_{i,j}^{\pi_i})}}_{(e)}}. \tag{49}$$

By applying the same argument as that of Equation (41) of Lemma D.8, we bound term $(e)$ with $SAL$.

We can rewrite term $(d)$ as follows:

$$(d) = \underbrace{\sum_{i=1}^{k} \sum_{j=h}^{H+1} V_{j+1}^{\pi_i}(x_{i,j}, a_{i,j}^{\pi_i})}_{(f)} + \underbrace{\sum_{i=1}^{k} \sum_{j=h}^{H-1} [\widehat{\mathbb{V}}_{i,j+1}(x_{i,j}, a_{i,j}^{\pi_i}) - \mathbb{V}_{j+1}^{\pi_i}(x_{i,j}, a_{i,j+1}^{\pi_i})]}_{(g)} \tag{50}$$

Under events $\mathcal{E}$ and $\Omega_{k,h}$, we can apply Lemmas D.5 and D.7 to upper bound terms $(f)$ and $(g)$ respectively, obtaining the following:

$$(f) \leqslant HT_k + 2\sqrt{H^4 T_k L} + \frac{4H^3 L}{3},$$

$$(g) \leqslant 7H^2 S\sqrt{AT_k L} + 2H \sum_{i=1}^{k} \sum_{j=h}^{H-1} \widetilde{\Delta}_{i,j+1}(x_{i,j+1}).$$

Plugging the bounds of $(f)$ and $(g)$ into Equation (50), we get:

$$(d) \leqslant HT_k + 2\sqrt{H^4 T_k L} + \frac{4H^3 L}{3} + 7H^2 S \sqrt{AT_k L} + 2H \sum_{i=1}^{k} \sum_{j=h}^{H-1} \widetilde{\Delta}_{i,j+1}(x_{i,j+1})$$

$$\leqslant 4HT_k + 2H \sum_{i=1}^{k} \sum_{j=h}^{H-1} \widetilde{\Delta}_{i,j+1}(x_{i,j+1}), \tag{51}$$

where Equation (51) holds under the condition of $[k]_{\text{typ}}$. Combining the bounds of terms $(d)$ and $(e)$, we can rewrite Equation (49) as:

$$(a) \leqslant \sqrt{4L} \sqrt{4HT_k + 2H \sum_{i=1}^{k} \sum_{j=h}^{H-1} \widetilde{\Delta}_{i,j+1}(x_{i,j+1}) \sqrt{SAL}}$$

$$\leqslant \sqrt{16HSAT_k L^2} + \sqrt{8HSAL^2 \sum_{i=1}^{k} \sum_{j=h}^{H-1} \widetilde{\Delta}_{i,j+1}(x_{i,j+1})}, \tag{52}$$

where Equation (52) is obtained by expanding the products and applying the subadditivity of the square root.

To bound term $(c)$, we apply Cauchy-Schwarz's inequality, obtaining:

$$(c) \leqslant 2 \sqrt{\underbrace{\sum_{i=1}^{k} \sum_{j=h}^{H-1} \mathbb{E}_{y \sim \widehat{P}_i(\cdot|x_{i,j}, a_{i,j}^{\pi_i})} b'_{i,j+1}(y)}_{(h)}} \sqrt{\underbrace{\sum_{i=1}^{k} \mathbb{I}(i \in [k]_{\text{typ}}) \sum_{j=h}^{H-1} \frac{1}{N_i(x_{i,j}, a_{i,j}^{\pi_i})}}_{(i)}}. \tag{53}$$

Similar to term $(e)$, we can bound term $(i)$ with $SAL$. We now bound term $(h)$. We can rewrite the term as:

$$(h) = \sum_{i=1}^{k} \sum_{j=h}^{H-1} \sum_{y \in \mathcal{S}} \widehat{P}_i(y|x_{i,j}, a_{i,j}^{\pi_i}) b'_{i,j+1}(y)$$

$$= \underbrace{\sum_{i=1}^{k} \sum_{j=h}^{H-1} \sum_{y \in \mathcal{S}} (\widehat{P}_i(y|x_{i,j}, a_{i,j}^{\pi_i}) - P(y|x_{i,j}, a_{i,j}^{\pi_i})) b'_{i,j+1}(y)}_{(j)} + \sum_{i=1}^{k} \sum_{j=h}^{H-1} \sum_{y \in \mathcal{S}} P(y|x_{i,j}, a_{i,j}^{\pi_i}) b'_{i,j+1}(y)$$

$$= (j) + \underbrace{\sum_{i=1}^{k} \sum_{j=h}^{H-1} \mathbb{E}_{y \sim P(\cdot|x_{i,j}, a_{i,j}^{\pi_i})} b'_{i,j+1}(y) - b'_{i,j+1}(x_{i,j+1})}_{(k)} + \underbrace{\sum_{i=1}^{k} \sum_{j=h}^{H-1} b'_{i,j=1}(x_{i,j+1})}_{(l)}. \tag{54}$$

We bound term $(j)$ as follows:

$$(j) \leqslant H^2 \sum_{i=1}^{k} \sum_{j=h}^{H-1} \|\widehat{P}_i(\cdot|x_{i,j}, a_{i,j}^{\pi_i}) - P(\cdot|x_{i,j}, a_{i,j}^{\pi_i})\|_1 \tag{55}$$

$$\leqslant 2H^2\sqrt{SL} \sum_{i=1}^{k} \sum_{j=h}^{H-1} (N_i(x_{i,j}, a_{i,j}^{\pi_i}))^{-1/2} \tag{56}$$

$$\leqslant 2H^2\sqrt{SL} \sum_{x \in \mathcal{S}} \sum_{a \in \mathcal{A}} \sum_{n=1}^{\frac{kH}{SA}} n^{-1/2}$$

$$\leqslant H^2 S \sqrt{AT_k L}, \tag{57}$$

where Equation (55) is obtained by bounding $b'_{i,j+1}(y)$ with $H^2$, Equation (56) follows by applying the result of Theorem 2.1 of (Weissman et al., 2003), which holds under event $\mathcal{E}$, and Equation (57) follows from a derivation similar to that of term $(a)$ of Lemma D.7.

To bound term $(k)$, we first observe that it is a Martingale difference sequence, and as such we can bound it via the event $\mathcal{E}_{\mathrm{az}}(\mathcal{F}_{b',k,h}, H^2, L)$, which holds under $\mathcal{E}$, obtaining:

$$(k) \leqslant 2H^2\sqrt{T_k L}.$$

By applying the definition of $b'$, we can bound term $(l)$ as:

$$(l) \leqslant 84^2 H^3 S^2 A L^2 \sum_{i=1}^{k} \sum_{j=h}^{H-1} \frac{1}{N'_{i,j+1}(x_{i,j+1})}$$

$$\leqslant 84^2 H^3 S^2 A L^2 \sum_{x \in \mathcal{S}} \sum_{n=1}^{T} n^{-1}$$

$$\leqslant 84^2 H^3 S^3 A L^3.$$

Plugging the bounds of terms $(j)$, $(k)$, and $(l)$ into Equation (54), we get:

$$(h) \leqslant H^2 S \sqrt{AT_k L} + 2H^2\sqrt{T_k L} + 84^2 H^3 S^3 A L^3.$$

By applying the bounds of terms $(h)$ and $(i)$ to Equation (53), we get:

$$(c) \leqslant 2\sqrt{H^2 S \sqrt{AT_k L} + 2H^2\sqrt{T_k L} + 84^2 H^3 S^3 A L^3} \sqrt{SAL}$$

$$\leqslant 2\sqrt{3HSAT_k L} + 2\sqrt{84^2 H^3 S^4 A^2 L^4}, \tag{58}$$

where Equation (58) is obtained by expanding the products, applying the subadditivity of the square root, and applying the definition of $[k]_{\mathrm{typ}}$.

Finally, we can combine the bounds of terms $(a)$, $(b)$, and $(c)$ into Equation (47), obtaining the following bound:

$$\sum_{i=1}^{k} \mathbb{I}(i \in [k]_{\text{typ}}) \sum_{j=h}^{H-1} b_{i,j}(x_{i,j}, a_{i,j}^{\pi_i})$$

$$\leqslant \sqrt{16HSAT_k L^2} + \sqrt{8HSAL^2 \sum_{i=1}^{k} \sum_{j=h}^{H-1} \widetilde{\Delta}_{i,j+1}(x_{i,j+1})}$$

$$+ \frac{7}{3}HSAL^2 + 2\sqrt{3HSAT_k L} + 2\sqrt{84^2 H^3 S^4 A^2 L^4}$$

$$\leqslant \sqrt{28HSAT_k L^2} + \frac{7}{3}HSAL^2$$

$$+ \sqrt{8HSAL^2 \sum_{i=1}^{k} \sum_{j=h}^{H-1} \widetilde{\Delta}_{i,j+1}(x_{i,j+1}) + 2\sqrt{84^2 H^3 S^4 A^2 L^4}},$$

thus demonstrating the result of Equation (45). By following the same procedure, we can obtain an upper bound to $\sum_{i=1}^{k} \mathbb{I}(i \in [k]_{\text{typ},x}, x_{i,h} = x) \sum_{j=h}^{H-1} b_{i,j}(x_{i,j}, a_{i,j}^{\pi_i})$ as:

$$\sum_{i=1}^{k} \mathbb{I}(i \in [k]_{\text{typ},x}, x_{i,h} = x) \sum_{j=h}^{H-1} b_{i,j}(x_{i,j}, a_{i,j}^{\pi_i})$$

$$\leqslant \sqrt{28H^2 SAN'_{k,h}(x)L^2} + \frac{7}{3}HSAL^2$$

$$+ \sqrt{8HSAL^2 \sum_{i=1}^{k} \mathbb{I}(x_{i,h} = x) \sum_{j=h}^{H-1} \widetilde{\Delta}_{i,j+1}(x_{i,j+1}) + 2\sqrt{84^2 H^3 S^4 A^2 L^4}},$$

thus concluding the proof. Following the same procedure, we can obtain the upper bound of Equation (46) by substituting terms $T_k$ with $HN'_{k,h}(x)$ and terms $\sum_{i=1}^{k} \sum_{j=h}^{H-1} \widetilde{\Delta}_{i,j}(x_{i,j})$ with $\sum_{i=1}^{k} \mathbb{I}(x_{i,h} = x) \sum_{j=h}^{H-1} \widetilde{\Delta}_{i,j}(x_{i,j})$.

$\square$

## E. Proof of Theorem 3.1

**Theorem 3.1** (Regret for `UCBVI` with Chernoff-Hoeffding bound)**.** *Let $\delta \in (0,1)$. Considering:*[9]

$$b_{k,h}(x,a) = \frac{2HL}{\sqrt{\max\{N_k(x,a), 1\}}},$$

*then, w.p. at least $1 - \delta$, the regret of* `UCBVI-CH` *is bounded by:*

$$\text{Reg}(\text{UCBVI-CH}, K) \leqslant 10eHL\sqrt{SAT} + \frac{8}{3}eH^2 S^2 AL^2,$$

*where $L = \ln\left(5HSAT/\delta\right)$. For $T \geqslant H^2 S^3 A$, this bound translates to $\widetilde{\mathcal{O}}(H\sqrt{SAT})$.*

We begin the proof by demonstrating optimism under the `UCBVI-CH` algorithm (i.e., every optimistic value function is an upper bound of the true optimal value function), which requires us to show that, with high probability, the event $\Omega := \{V_{k,h}(x) \geqslant V_h^*(x), \forall k \in \llbracket K \rrbracket, h \in \llbracket H \rrbracket, x \in \mathcal{S}\}$.

**Lemma E.1** (Optimism under Chernoff-Hoeffding bonus)**.** *Let the optimistic bonus be defined as:*

$$b_{k,h}(x,a) = \frac{2HL}{\sqrt{N_k(x,a)}}.$$

---

[9]We assume that, by definition, $b_{k,H}(s,a) = 0$, as at the last stage there is no need for exploration and the rewards are deterministic.

*Then, under event $\mathcal{E}$, the following event holds:*

$$\Omega := \{V_{k,h}(x) \geqslant V_h^*(x), \forall k \in [\![K]\!], h \in [\![H]\!], x \in \mathcal{S}\}.$$

*Proof.* We demonstrate the result by induction. Let $V_{k,h}$ be the optimistic value function at stage $h$ computed using the history up to the end of episode $k - 1$, and let $V_h^*$ be the true optimal value function at stage $h$.

By definition, $V_{k,H+1}(x) = V_{H+1}^*(x) = 0$ for every $x \in \mathcal{S}$, and thus the inequality $V_{k,H+1} \geqslant V_{H+1}^*$ trivially holds. To prove the inductive step, we need to demonstrate that, if $V_{k,h+1} \geqslant V_{h+1}^*$ holds, then it also holds that $V_{k,h} \geqslant V_h^*$. We can drive this result as follows:

$$
\begin{aligned}
V_{k,h}(x) - V_h^* &= \max_{a \in \mathcal{A}} Q_{k,h}(x, a) - V_h^*(x) \\
&\geqslant Q_{k,h}(x, a_{k,h}^{\pi^*}) - V_h^*(x) \\
&= \sum_{y \in \mathcal{S}} \widehat{P}_k(y|x, a_{k,h}^{\pi^*}) V_{k,h+1}(y) + b_{k,h}(x, a_{k,h}^{\pi^*}) - \sum_{y \in \mathcal{S}} P(y|x, a_{k,h}^{\pi^*}) V_{h+1}^*(y) \\
&\geqslant \sum_{y \in \mathcal{S}} \left[ \widehat{P}_k(y|x, a_{k,h}^{\pi^*}) - P(y|x, a_{k,h}^{\pi^*}) V_{h+1}^*(y) \right] + b_{k,h}(x, a_{k,h}^{\pi^*}) &&(59) \\
&\geqslant b_{k,h}(x, a_{k,h}^{\pi^*}) - 2\sqrt{\frac{H^2 L}{N_k(x_{k,h}, a_{k,h}^{\pi^*})}} &&(60) \\
&\geqslant 2\sqrt{\frac{H^2 L}{N_k(x_{k,h}, a_{k,h}^{\pi^*})}} - 2\sqrt{\frac{H^2 L}{N_k(x_{k,h}, a_{k,h}^{\pi^*})}} \\
&\geqslant 0,
\end{aligned}
$$

where Equation (59) follows by the inductive hypothesis, Equation (60) is obtained because, under $\mathcal{E}$, we can bound $|\widehat{P}_k(y|x, a_{k,h}^{\pi^*}) - P(y|x, a_{k,h}^{\pi^*}) V_{h+1}^*(y)|$ by applying Azuma-Hoeffding's inequality, allowing us to simplify terms and show optimism. $\qquad\square$

Our objective is to bound the regret after $K$ episodes (i.e., $\mathrm{Reg}(\texttt{UCBVI-CH}, K)$). We can observe that, under event $\Omega$, it holds that:

$$
\begin{aligned}
\mathrm{Reg}(\texttt{UCBVI-CH}, K) &= \sum_{k \in [\![K]\!]} V_1^*(x_{k,1}) - V_1^{\pi_k}(x_{k,1}) \\
&\leqslant \sum_{k \in [\![K]\!]} V_{k,1}(x_{k,1}) - V_1^{\pi_k}(x_{k,1}) \\
&= \sum_{k \in [\![K]\!]} \widetilde{\Delta}_{k,1}(x_{k,1}) \\
&= \widetilde{\mathrm{Reg}}(\texttt{UCBVI-CH}, K).
\end{aligned}
$$

As such, we can now focus on finding an upper bound to $\widetilde{\mathrm{Reg}}(\texttt{UCBVI-CH}, K)$. By applying Lemma D.2, we can write:

$$
\begin{aligned}
\widetilde{\mathrm{Reg}}(\texttt{UCBVI-CH}, K) &= \sum_{k \in [\![K]\!]} \widetilde{\Delta}_{k,1}(x_{k,1}) \\
&\leqslant e \sum_{i=1}^{K} \sum_{j=1}^{H-1} \left[ \varepsilon_{i,j} + 2\sqrt{L}\bar{\varepsilon}_{i,j} + b_{i,j}(x_{i,j}, a_{i,j}^{\pi_i}) + \xi_{i,j}(x_{i,j}, a_{i,j}^{\pi_i}) + \frac{8H^2 SL}{3N_i(x_{i,j}, a_{i,j}^{\pi_i})} \right]. &&(61)
\end{aligned}
$$

To find an upper bound to the regret, we can thus bound the summation of each of the terms individually.

By applying Lemma D.3, we obtain the following bounds:

$$\sum_{i=1}^{K} \sum_{j=1}^{H-1} \varepsilon_{i,j} \leqslant 2\sqrt{H^2 T L},$$

$$\sum_{i=1}^{K} \sum_{j=1}^{H-1} 2\sqrt{L} \bar{\varepsilon}_{i,j} \leqslant 4\sqrt{T L}.$$

Then, we can derive the following bound:

$$\sum_{i=1}^{K} \sum_{j=1}^{H} \frac{8H^2 S L}{3 N_i(x_{i,j}, a_{i,j}^{\pi_i})} = \frac{8}{3} H^2 S L \sum_{x \in \mathcal{S}} \sum_{a \in \mathcal{A}} \sum_{n=1}^{N_K(x,a)} n^{-1} \tag{62}$$

$$\leqslant \frac{8}{3} H^2 S L \sum_{x \in \mathcal{S}} \sum_{a \in \mathcal{A}} \sum_{n=1}^{\frac{KH}{SA}} n^{-1} \tag{63}$$

$$\leqslant \frac{8}{3} H^2 S^2 A L^2$$

where Equation (62) is obtained by rearranging the terms to isolate the summation of $n^{-1}$ for $n$ from 1 to $N_K(x, a)$ (i.e., the total number of times each state-action pair has been observed up to the end of episode $K$), and Equation (63) derives from the observation that the summation can be upper bounded by considering a uniform state-action visit distribution. This derivation produces the same result as applying the well-known pigeonhole principle.

By applying a similar reasoning, we bound the remaining summations over the bonus terms:

$$\sum_{i=1}^{K} \sum_{j=1}^{H} b_{i,j}(x_{i,j}) = \sum_{i=1}^{K} \sum_{j=1}^{H} 2H \sqrt{\frac{L}{N_i(x_{i,j}, a_{i,j}^{\pi_i})}}$$

$$= 2H\sqrt{L} \sum_{x \in \mathcal{S}} \sum_{a \in \mathcal{A}} \sum_{n=1}^{N_K(x,a)} n^{-1/2}$$

$$\leqslant 2H\sqrt{L} \sum_{x \in \mathcal{S}} \sum_{a \in \mathcal{A}} \sum_{n=1}^{\frac{KH}{SA}} n^{-1/2}$$

$$\leqslant 2\sqrt{H^2 S A T L},$$

and over the model error terms:

$$\sum_{i=1}^{K} \sum_{j=1}^{H} \xi_{i,j}(x_{i,j}, a_{i,j}^{\pi_i}) \leqslant \sum_{i=1}^{K} \sum_{j=1}^{H} 2H \sqrt{\frac{L}{N_i(x_{i,j}, a_{i,j}^{\pi_i})}} \tag{64}$$

$$= 2H\sqrt{L} \sum_{x\in\mathcal{S}} \sum_{a\in\mathcal{A}} \sum_{n=1}^{N_K(x,a)} n^{-1/2}$$

$$\leqslant 2HL \sum_{x\in\mathcal{S}} \sum_{a\in\mathcal{A}} \sum_{n=1}^{\frac{KH}{SA}} n^{-1/2}$$

$$\leqslant 2\sqrt{H^2 SATL},$$

where Equation (64) is obtained by bounding $\xi_{i,j}(x_{i,j}, a_{i,j}^{\pi_i})$ using the Chernoff-Hoeffding inequality. Finally, we can put all the bounds together and rewrite Equation (61) as:

$$\widetilde{\text{Reg}}(\texttt{UCBVI-CH}, K) \leqslant e\left[ 2\sqrt{H^2 TL} + 4\sqrt{TL} + 2\sqrt{H^2 SATL} + 2\sqrt{H^2 SATL} + \frac{8}{3} H^2 S^2 AL^2 \right]$$

$$\leqslant e\left[ 10\sqrt{H^2 SATL} + \frac{8}{3} H^2 S^2 AL^2 \right],$$

thus completing the proof.

## F. Proof of Theorem 3.2

**Theorem 3.2** (Regret for `UCBVI` with Bernstein-Freedman bound)**.** *Let $\delta \in (0, 1)$. Considering:*[5]

$$b_{k,h}(x,a) = \underbrace{\sqrt{\frac{4L\,\mathbb{V}\text{ar}_{y\sim\hat{P}_k(\cdot|x,a)}(V_{k,h+1}(y))}{\max\{N_k(x,a), 1\}}}}_{\text{(A)}} +$$

$$+ \underbrace{\frac{7HL}{3\max\{N_k(x,a)-1, 1\}}}_{\text{(B)}} +$$

$$+ \underbrace{\sqrt{\frac{4\sum_{y\in\mathcal{S}}\left(\hat{P}(y|x,a)\cdot\min\left\{\frac{84^2 H^3 S^2 AL^2}{\max\{1, N'_{k,h+1}(y)\}}, H^2\right\}\right)}{\max\{N_k(x,a), 1\}}}}_{\text{(C)}},$$

*then, w.p. at least $1 - \delta$, the regret of `UCBVI-BF-I` is bounded by:*

$$\text{Reg}(\texttt{UCBVI-BF-I}, K) \leqslant 24eL\sqrt{HSAT} +$$

$$+ 616eH^2 S^2 AL^2 + 4e\sqrt{H^2 TL},$$

*where $L = \ln(5HSAT/\delta)$. For $T \geqslant H^3 S^3 A$ and $SA \geqslant H$, this bound translates to $\widetilde{\mathcal{O}}(\sqrt{HSAT})$.*

Similarly to the proof of Theorem 3.1 in Appendix E, in order to demonstrate the upper bound of `UCBVI-BF-I`, we first need to demonstrate optimism. However, in order to remove the additional $\sqrt{H}$ term, we are required to both demonstrate optimism as well as to bound by how much the optimistic value function estimator exceeds the true optimal value function.

We start by observing that:

$$\text{Reg}(\texttt{UCBVI-BF-I}, K) = \sum_{k \in [\![K]\!]} V_1^*(x_{k,1}) - V_1^{\pi_k}(x_{k,1})$$

$$\leqslant \sum_{k \in [\![K]\!]} V_{k,1}(x_{k,1}) - V_1^{\pi_k}(x_{k,1})$$

$$= \sum_{k \in [\![K]\!]} \widetilde{\Delta}_{k,1}(x_{k,1})$$

$$= \widetilde{\text{Reg}}(\texttt{UCBVI-BF-I}, K).$$

According to Lemma D.2, under the events $\mathcal{E}$ and $\Omega_{k,h}$, we can decompose the pseudo-regret as:

$$\sum_{i=1}^{k} \widetilde{\Delta}_{i,h}(x_{i,h}) \leqslant e \sum_{i=1}^{k} \sum_{j=h}^{H-1} \left[ \varepsilon_{i,j} + 2\sqrt{L}\bar{\varepsilon}_{i,j} + b_{i,j}(x_{i,j}, a_{i,j}^{\pi_i}) + \xi_{i,j}(x_{i,j}, a_{i,j}^{\pi_i}) + \frac{8H^2 SL}{3N_i(x_{i,j}, a_{i,j}^{\pi_i})} \right]. \tag{65}$$

We also define, by trivially modifying the derivation of Lemma D.2, the pseudo-regret considering only the episodes in which, at stage $h \in [\![H]\!]$ a specific state $x \in \mathcal{S}$ was occupied:

$$\sum_{i=1}^{k} \mathbb{I}(x_{i,h} = x)\Delta_{i,h}(x_{i,h}) \leqslant \sum_{i=1}^{k} \mathbb{I}(x_{i,h} = x)\widetilde{\Delta}_{i,h}(x_{i,h})$$

$$\leqslant e \sum_{i=1}^{k} \mathbb{I}(x_{i,h} = x) \sum_{j=h}^{H-1} \left[ \varepsilon_{i,j} + 2\sqrt{L}\bar{\varepsilon}_{i,j} + b_{i,j}(x_{i,j}, a_{i,j}^{\pi_i}) \right.$$

$$\left. + \xi_{i,j}(x_{i,j}, a_{i,j}^{\pi_i}) + \frac{8H^2 SL}{3N_i(x_{i,j}, a_{i,j}^{\pi_i})} \right]. \tag{66}$$

By applying Lemmas D.3 and D.4, we can upper bound Equations (65) and (66) as:

$$\sum_{i=1}^{k} \widetilde{\Delta}_{i,h}(x_{i,h}) \leqslant e \sum_{i=1}^{k} \sum_{j=h}^{H-1} \left[ b_{i,j}(x_{i,j}, a_{i,j}^{\pi_i}) + \xi_{i,j}(x_{i,j}, a_{i,j}^{\pi_i}) + \frac{8H^2 SL}{3N_i(x_{i,j}, a_{i,j}^{\pi_i})} \right]$$

$$+ 2e\sqrt{H^2 T_k L} + 4e\sqrt{T_k L}$$

$$= U_{k,h},$$

and:

$$\sum_{i=1}^{k} \mathbb{I}(x_{i,h} = x)\widetilde{\Delta}_{i,h}(x_{i,h}) \leqslant e \sum_{i=1}^{k} \mathbb{I}(x_{i,h} = x) \sum_{j=h}^{H-1} \left[ b_{i,j}(x_{i,j}, a_{i,j}^{\pi_i}) + \xi_{i,j}(x_{i,j}, a_{i,j}^{\pi_i}) + \frac{8H^2 SL}{3N_i(x_{i,j}, a_{i,j}^{\pi_i})} \right]$$

$$+ 2e\sqrt{H^3 N_{k,h}'(x)L} + 4e\sqrt{H N_{k,h}'(x)L}$$

$$= U_{k,h,x},$$

where we denote the upper bounds of $\sum_{i=1}^{k} \widetilde{\Delta}_{i,h}(x_{i,h})$ and $\sum_{i=1}^{k} \mathbb{I}(x_{i,h} = x)\widetilde{\Delta}_{i,h}(x_{i,h})$ as $U_{k,h}$ and $U_{k,h,x}$, respectively, for ease of notation.

We now demonstrate optimism, which requires us to show that, with high probability, the event $\Omega_{k,h}$ holds.

**Lemma F.1** (Optimism under Bernstein-Freedman bonus). *Let the optimistic bonus be defined as:*

$$b_{k,h}(x,a) = \sqrt{\frac{4L\,\mathbb{V}\mathrm{ar}_{y\sim\hat{P}_k(\cdot|x,a)}[V_{k,h+1}(y)]}{N_k(x,a)}} + \frac{7HL}{3(N_k(x,a)-1)}$$

$$+ \sqrt{\frac{4\min\{\mathbb{E}_{y\sim\hat{P}_k(\cdot|x,a)}[\frac{84^2H^3S^2AL^2}{N'_{k,h+1}(y)}],H^2\}}{N_k(x,a)}}.$$

*Then, under event $\mathcal{E}$, the following set of events hold:*

$$\Omega_{k,h} := \left\{V_{i,j}(x) \geqslant V_j^*(x), \forall (i,j) \in [k,h]_{\mathrm{hist}}, x \in \mathcal{S}\right\},$$

*for $k \in [\![K]\!]$ and $h \in [\![H]\!]$, where:*

$$[k,h]_{\mathrm{hist}} := \{(i,j) \in [\![K]\!] \times [\![H]\!] : i < k \vee (i = h, j \geqslant h)\}.$$

*Proof.* We demonstrate the result by induction. We begin by observing that $V_{k,H+1}(x) = V_{H+1}^*(x) = 0$ for every $k \in [\![K]\!]$ and $x \in \mathcal{S}$. To prove the induction, we need to prove that, if $\Omega_{k,h}$ holds, then also $\Omega_{k,h-1}$ holds. We prove this for a generic $k \in [\![K]\!]$, and we can then apply this procedure for increasing values of $k$, starting from $k = 1$.

If $\Omega_{k,h}$ holds, then $V_{k,h}(x) \geqslant V_h^*(x)$ for every $x \in \mathcal{S}$. We now bound the estimation error due to the optimistic approach:

$$V_{k,h}(x) - V_h^*(x) = \frac{1}{N'_{k,h}(x)} \sum_{i=1}^{k} \mathbb{I}(x_{i,h} = x)(V_{k,h}(x) - V_h^*(x))$$

$$\leqslant \frac{1}{N'_{k,h}(x)} \sum_{i=1}^{k} \mathbb{I}(x_{i,h} = x)(V_{i,h}(x) - V_h^{\pi_i}(x)) \tag{67}$$

$$= \frac{1}{N'_{k,h}(x)} \sum_{i=1}^{k} \mathbb{I}(x_{i,h} = x)\widetilde{\Delta}_{i,h}(x_{i,h}), \tag{68}$$

where Equation (67) follows from the fact that $V_{k,h}$ is monotonically decreasing in $k$ by definition, and by observing that $V_h^* \geqslant V_h^{\pi_i}$.

Recalling the upper bound of $\sum_{i=1}^{k} \mathbb{I}(x_{i,h} = x)\widetilde{\Delta}_{i,h}(x_{i,h})$:

$$\sum_{i=1}^{k} \mathbb{I}(x_{i,h} = x)\widetilde{\Delta}_{i,h}(x_{i,h}) \leqslant U_{k,h,x}$$

$$= e \sum_{i=1}^{k} \mathbb{I}(x_{i,h} = x) \sum_{j=h}^{H-1} \left[ b_{i,j}(x_{i,j}, a_{i,j}^{\pi_i}) + \xi_{i,j}(x_{i,j}, a_{i,j}^{\pi_i}) \right.$$

$$\left. + \frac{8H^2SL}{3N_i(x_{i,j}, a_{i,j}^{\pi_i})} \right] + 2e\sqrt{H^3 N'_{k,h}(x)L} + 4e\sqrt{HN'_{k,h}(x)L},$$

we now bound the summations over the terms in the summation over episodes and stages. By applying Lemma D.9, we can bound the summation over typical episodes of the bonus terms as:

$$\sum_{i=1}^{k} \mathbb{I}(i \in [k]_{\text{typ},x}, x_{i,h} = x) \sum_{j=h}^{H-1} b_{i,j}(x_{i,j}, a_{i,j}^{\pi_i})$$

$$\leqslant \sqrt{28H^2 SAN'_{k,h}(x)L^2} + \frac{7}{3}HSAL^2 + 2\sqrt{84^2 H^3 S^4 A^2 L^4}$$

$$+ \sqrt{8H^2 SAL^2 U_{k,h,x}},$$

by observing that $\sum_{i=1}^{k} \mathbb{I}(x_{i,h} = x)\Delta_{i,h}(x_{i,h}) \leqslant U_{k,h,x}$ and that the series of $U_{k,h,x}$ terms is decreasing in $h$, as each term $U_{k,h,x}$ is a summation of elements which includes the next term, and as such we can upper bound $\sum_{j=h}^{H-1} U_{k,j,x}$ with $HU_{k,h,x}$.

In a similar way, we can apply the result of Lemma D.8 to bound the summation over typical episodes of the state-action wise model error terms as:

$$\sum_{i=1}^{k} \mathbb{I}(i \in [k]_{\text{typ},x}, x_{i,h} = x) \sum_{j=h}^{H-1} \left[ \widehat{P}_i(\cdot|x_{i,j}, a_{i,j}^{\pi_i}) - P(\cdot|x_{i,j}, a_{i,j}^{\pi_i}) \right]^\top V_{j+1}^*(\cdot)$$

$$\leqslant \sqrt{6H^2 SAN'_{k,h}(x)L^2} + \frac{2}{3}HSAL^2$$

$$+ 2\sqrt{H^2 SAL^2 U_{k,h,x}}.$$

With the same procedure as in the proof of Lemma E.1, we obtain the following upper bound:

$$\sum_{i=1}^{K} \sum_{j=1}^{H} \frac{8H^2 SL}{3N_i(x_{i,j}, a_{i,j}^{\pi_i})} \leqslant \frac{8}{3}H^2 S^2 AL^2. \tag{69}$$

By combining these result, and accounting for the regret on non-typical episodes, we can write:

$$\sum_{i=1}^{k} \mathbb{I}(x_{i,h} = x)\Delta_{i,h}(x_{i,h}) \leqslant U_{k,h,x}$$

$$\leqslant e\Bigg[ \sqrt{28H^2 SAN'_{k,h}(x)L^2} + \frac{7}{3}HSAL^2 + 168\sqrt{H^3 S^4 A^2 L^4}$$

$$+ \sqrt{8H^2 SAL^2 U_{k,h,x}} + \sqrt{6H^2 SAN'_{k,h}(x)L^2} + \frac{2}{3}HSAL^2$$

$$+ 2\sqrt{H^2 SAL^2 U_{k,h,x}} + \frac{8}{3}H^2 S^2 AL^2 + 2\sqrt{H^3 N'_{k,h}(x)L}$$

$$+ 4\sqrt{HN'_{k,h}(x)L} + 100H^2 S^2 AL^2 \Bigg]$$

$$\leqslant e\Bigg[ 12\sqrt{H^2 SAN'_{k,h}(x)L^2} + 5\sqrt{H^2 SAL^2 U_{k,h,x}}$$

$$+ \frac{821}{3}H^2 S^2 AL^2 + 2\sqrt{H^3 N'_{k,h}(x)L} \Bigg].$$

Letting:

$$\alpha = e\left[12\sqrt{H^2 SAN'_{k,h}(x)L^2} + \frac{821}{3}H^2 S^2 AL^2 + 2\sqrt{H^3 N'_{k,h}(x)L}\right],$$
$$\beta = 5e\sqrt{H^2 S^2 AL^2},$$

we can solve for $U_{k,h,x}$ and obtain the following upper bound:

$$U_{k,h,x} \leqslant \beta^2 + 2\alpha,$$

which we can write as:

$$
\begin{aligned}
U_{k,h,x} &\leqslant 25e^2 H^2 S^2 AL^2 + 24e\sqrt{H^2 SAN'_{k,h}(x)L^2} + \frac{1642}{3}eH^2 S^2 AL^2 + 4e\sqrt{H^3 N'_{k,h}(x)L} \\
&\leqslant 24e\sqrt{H^2 SAN'_{k,h}(x)L^2} + \frac{1846}{3}eH^2 S^2 AL^2 + 4e\sqrt{H^3 N'_{k,h}(x)L} \\
&\leqslant 28e\sqrt{H^2 SAN'_{k,h}(x)L^2} + \frac{1846}{3}eH^2 S^2 AL^2 \tag{70} \\
&\leqslant 28 \cdot \frac{12}{11}e\sqrt{H^3 S^2 AN'_{k,h}(x)L^2} \tag{71} \\
&\leqslant 84\sqrt{H^3 S^2 AN'_{k,h}(x)L^2},
\end{aligned}
$$

where Equation (70) holds if $SA \geqslant H$, and Equations (71) holds under the condition of $[k]_{\text{typ},x}$.

Plugging this result into Equation (68), and observing that the error cannot be greater than $H$, we get the following upper bound to the estimation error due to the optimistic approach:

$$V_{k,h}(x) - V_h^*(x) \leqslant \min\left\{84\sqrt{\frac{H^3 S^2 AL^2}{N'_{k,h}(x)}}, H\right\}. \tag{72}$$

Using this result, we now prove that $V_{k,h-1}(x) \geqslant V_{h-1}^*(x)$. Let us recall the definition of $V_{k,h-1}(x)$:

$$V_{k,h-1}(x) = \min\left\{V_{k-1,h-1}(x), H, \mathcal{T}_{h-1}^{\pi_k} V_{k,h}\right\},$$

where $\mathcal{T}_{h-1}^{\pi_k} V_{k,h} := R^{\pi_k}(x_{k,h-1}) + b_{k,h-1}(x_{k,h-1}, a_{k,h-1}^{\pi_k}) + \mathbb{E}_{y \sim \hat{P}_k(\cdot|x_{k,h-1}, a_{k,h-1}^{\pi_k})} V_{k,h}(y)$. Observe that, if $V_{k,h-1}(x) = H$, the optimism holds trivially. Also, if $V_{k,h-1}(x) = V_{k-1,h-1}(x)$, the optimism hold trivially under $\Omega_{k,h}$. As such, we only need to demonstrate the case in which $V_{k,h-1}(x) = \mathcal{T}_{h-1}^{\pi_k} V_{k,h}$. As such, we derive the following:

$$
\begin{aligned}
V_{k,h-1}(x) - V_{h-1}^*(x) &= \max_{a \in \mathcal{A}} \left\{ R(x,a) + b_{k,h-1}(x,a) + \sum_{y \in \mathcal{S}} \widehat{P}_k(y|x,a) V_{k,h}(y) \right\} \\
&\quad - R(x, a_{k,h-1}^{\pi^*}) - \sum_{y \in \mathcal{S}} P(y|x, a_{k,h-1}^{\pi^*}) V_h^*(y) \\
&\geqslant b_{k,h-1}(x, a_{k,h-1}^{\pi^*}) + \sum_{y \in \mathcal{S}} \widehat{P}_k(y|x, a_{k,h-1}^{\pi^*}) V_{k,h}(y) \\
&\quad - \sum_{y \in \mathcal{S}} P(y|x, a_{k,h-1}^{\pi^*}) V_h^*(y) \\
&= b_{k,h-1}(x, a_{k,h-1}^{\pi^*}) + \sum_{y \in \mathcal{S}} \widehat{P}_k(y|x, a_{k,h-1}^{\pi^*}) \left[ V_{k,h}(y) - V_h^*(y) \right] \\
&\quad + \sum_{y \in \mathcal{S}} \left[ \widehat{P}_k(y|x, a_{k,h-1}^{\pi^*}) - P(y|x, a_{k,h-1}^{\pi^*}) \right] V_h^*(y) \\
&\geqslant b_{k,h-1}(x, a_{k,h-1}^{\pi^*}) + \sum_{y \in \mathcal{S}} \widehat{P}_k(y|x, a_{k,h-1}^{\pi^*}) \left[ V_{k,h}(y) - V_h^*(y) \right] \qquad (73)
\end{aligned}
$$

where Equation (73) follows from the induction assumption.

Under event $\mathcal{E}$, we can apply the empirical Bernstein inequality (Maurer & Pontil, 2009):

$$
\left| \sum_{y \in \mathcal{S}} \left[ \widehat{P}_k(y|x,a) - P(y|x,a) \right] V_h^*(y) \right| \leqslant \sqrt{\frac{2 \widehat{\mathbb{V}}_{k,h}^*(x,a) L}{N_k(x,a)}} + \frac{7HL}{3(N_k(x,a) - 1)},
$$

where $\widehat{\mathbb{V}}_{k,h}^*(x,a) := \mathbb{V}\mathrm{ar}_{y \sim \widehat{P}_k(\cdot|x,a)}[V_h^*(y)]$. As such, we obtain:

$$
\begin{aligned}
V_{k,h-1}(x) - V_{h-1}^*(x) &\geqslant b_{k,h-1}(x, a_{k,h-1}^{\pi^*}) - \sqrt{\frac{2 \widehat{\mathbb{V}}_{k,h}^*(x, a_{k,h-1}^{\pi^*}) L}{N_k(x, a_{k,h-1}^{\pi^*})}} - \frac{7HL}{3(N_k(x, a_{k,h-1}^{\pi^*}) - 1)} \\
&= \sqrt{\frac{4 \widehat{\mathbb{V}}_{k,h}(x, a_{k,h-1}^{\pi^*}) L}{N_k(x, a_{k,h-1}^{\pi^*})}} + \sqrt{\frac{4 L \mathbb{E}_{y \sim \widehat{P}_k(\cdot|x, a_{k,h-1}^{\pi^*})} b'_{k,h}(y)}{N_k(x, a_{k,h-1}^{\pi^*})}} \\
&\quad - \sqrt{\frac{2 \widehat{\mathbb{V}}_{k,h}^*(x, a_{k,h-1}^{\pi^*}) L}{N_k(x, a_{k,h-1}^{\pi^*})}}. \qquad (74)
\end{aligned}
$$

We now bound $\widehat{\mathbb{V}}_{k,h}^*$ in terms of $\widehat{\mathbb{V}}_{k,h}$. Observing that:

$$
\begin{aligned}
\mathbb{V}\mathrm{ar}[X] &= \mathbb{E}[X - \mathbb{E}[X]]^2 \\
&= \mathbb{E}[X \pm Y - \mathbb{E}[X] \pm \mathbb{E}[Y]]^2 \\
&= \mathbb{E}[(X - Y) - \mathbb{E}[X - Y] + Y - \mathbb{E}[Y]]^2 \\
&\leqslant 2\mathbb{E}[(X - Y) - \mathbb{E}[X - Y]]^2 + 2\mathbb{E}[Y - \mathbb{E}[Y]]^2 \\
&= \mathbb{V}\mathrm{ar}[X - Y] + 2\mathbb{V}\mathrm{ar}[Y],
\end{aligned}
$$

we can then rewrite:

$$\widehat{\mathbb{V}}_{k,h}^*(x, a_{k,h-1}^{\pi^*}) \leqslant 2\widehat{\mathbb{V}}_{k,h}(x, a_{k,h-1}^{\pi^*}) + 2 \operatorname*{\mathbb{V}ar}_{y \sim \widehat{P}_k(\cdot|x, a_{k,h-1}^{\pi^*})} [V_h^*(y) - V_{k,h}(y)]$$

$$\leqslant 2\widehat{\mathbb{V}}_{k,h}(x, a_{k,h-1}^{\pi^*}) + 2\sum_{y \in \mathcal{S}} \widehat{P}_k(\cdot|x, a_{k,h-1}^{\pi^*})(V_{k,h}(y) - V_y^*)^2.$$

By plugging this result into Equation (74), we get:

$$V_{k,h-1}(x) - V_{h-1}^*(x) \geqslant \sqrt{\frac{4L\mathbb{E}_{y \sim \widehat{P}_k(\cdot|x, a_{k,h-1}^{\pi^*})} b_{k,h}'(y)}{N_k(x, a_{k,h-1}^{\pi^*})}} - \sqrt{\frac{4L\mathbb{E}_{y \sim \widehat{P}_k(\cdot|x, a_{k,h-1}^{\pi^*})}(V_{k,h}(y) - V_h^*(y))^2}{N_k(x, a_{k,h-1}^{\pi^*})}}.$$

By applying the result of Equation (72) and the definition of $b_{k,h}'(y)$, we finally obtain that $V_{k,h-1}(x) - V_{h-1}^*(x) \geqslant 0$, thus demonstrating optimism. $\qquad\square$

Having demonstrated optimism, we now prove the upper bound of the regret $\mathrm{Reg}(\mathtt{UCBVI-BF-I}, K)$:

$$\begin{aligned}
\widetilde{\mathrm{Reg}}(\mathtt{UCBVI-BF-I}, K) &\leqslant U_{K,1} \\
&= e\Bigg[ \sqrt{28HSATL^2} + \frac{7}{3}HSAL^2 + 2\sqrt{84^2 H^3 S^4 A^2 L^4} \\
&\quad + \sqrt{8HSAL^2 U_{K1}} + \sqrt{6HSATL^2} + \frac{2}{3}HSAL^2 \\
&\quad + 2\sqrt{HSAL^2 U_{K,1}} + \frac{8}{3}H^2 S^2 AL^2 + 2\sqrt{H^2 TL} \\
&\quad + 4\sqrt{TL} + 100H^2 S^2 AL \Bigg] \\
&\leqslant e\Bigg[ 12\sqrt{HSATL^2} + 5\sqrt{H^2 SAL^2 U_{K,1}} + \frac{821}{3}H^2 S^2 AL^2 + 2\sqrt{H^2 TL} \Bigg]
\end{aligned} \tag{75}$$

where Equation (75) is obtained by applying the results of Lemmas D.9 and D.8, by applying the result of Equation (69), and by accounting for the regret of non-typical episodes.

As done in Lemma F.1, by letting:

$$\begin{aligned}
\alpha &= e\Bigg[ 12\sqrt{HSATL^2} + \frac{821}{3}H^2 S^2 AL^2 + 2\sqrt{H^2 TL} \Bigg], \\
\beta &= 5e\sqrt{H^2 S^2 AL^2},
\end{aligned}$$

we can solve for $U_{K,1}$ and obtain:

$$\begin{aligned}
\widetilde{\mathrm{Reg}}(\mathtt{UCBVI-BF-I}, K) &\leqslant 24e\sqrt{HSATL^2} + \frac{1846}{3}eH^2 S^2 AL^2 + 4e\sqrt{H^2 TL} \\
&\leqslant 24e\sqrt{HSATL^2} + 616eH^2 S^2 AL^2 + 4e\sqrt{H^2 TL}
\end{aligned}$$

thus completing the proof.

# G. Additional Illustrative Experiments

In this section, we report additional experiments for the illustrative environment presented in Section 4.1, comparing the UCBVI-BF, UCBVI-BF-I, and MVP algorithms in the case of larger state and action spaces. In particular, we compare the three algorithms in environments with state and action spaces with cardinalities $S, A \in \{3, 5, 10\}$, time horizon $H = 5$ (Figure 4) and $H = 10$ (Figure 5), and number of episodes $K = 10^5$. The results are averaged over 5 runs, with a 95% confidence interval.

As we can see from the figures, both UCBVI-BF and UCBVI-BF-I outperform MVP in all the evaluated settings. Additionally, we can observe that the performance gap between UCBVI-BF and UCBVI-BF-I increases with the size of the environment, showing that the reduction in unnecessary exploration of UCBVI-BF-I provides a greater performance improvement in larger environments.

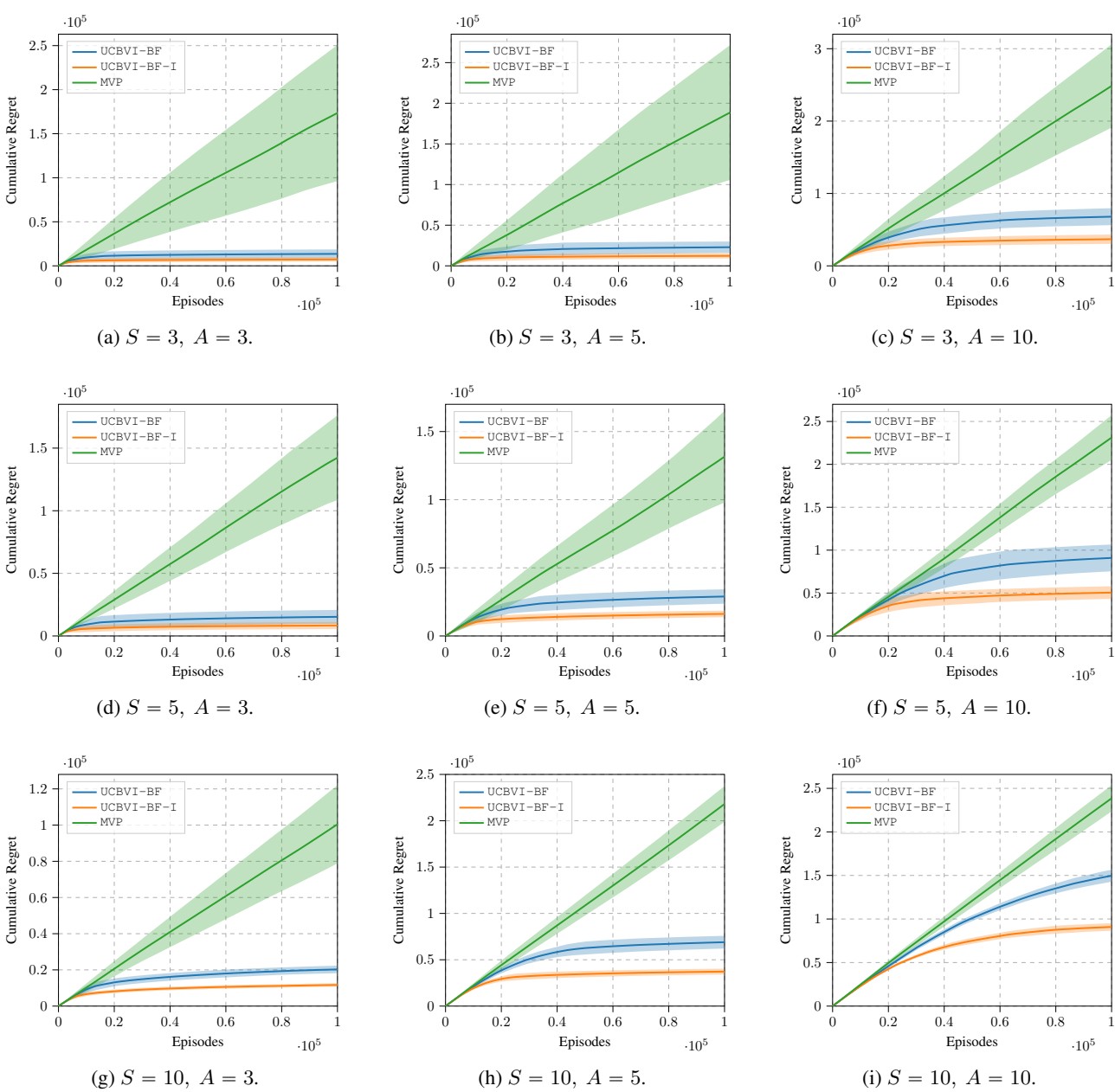

*Figure 4.* Performances in terms of cumulative regret in toy environments with $S \in \{3, 5, 10\}$ states and $A \in \{3, 5, 10\}$ actions for $H = 5$ and $K = 10^5$ (5 runs, mean $\pm$ 95% C.I.).

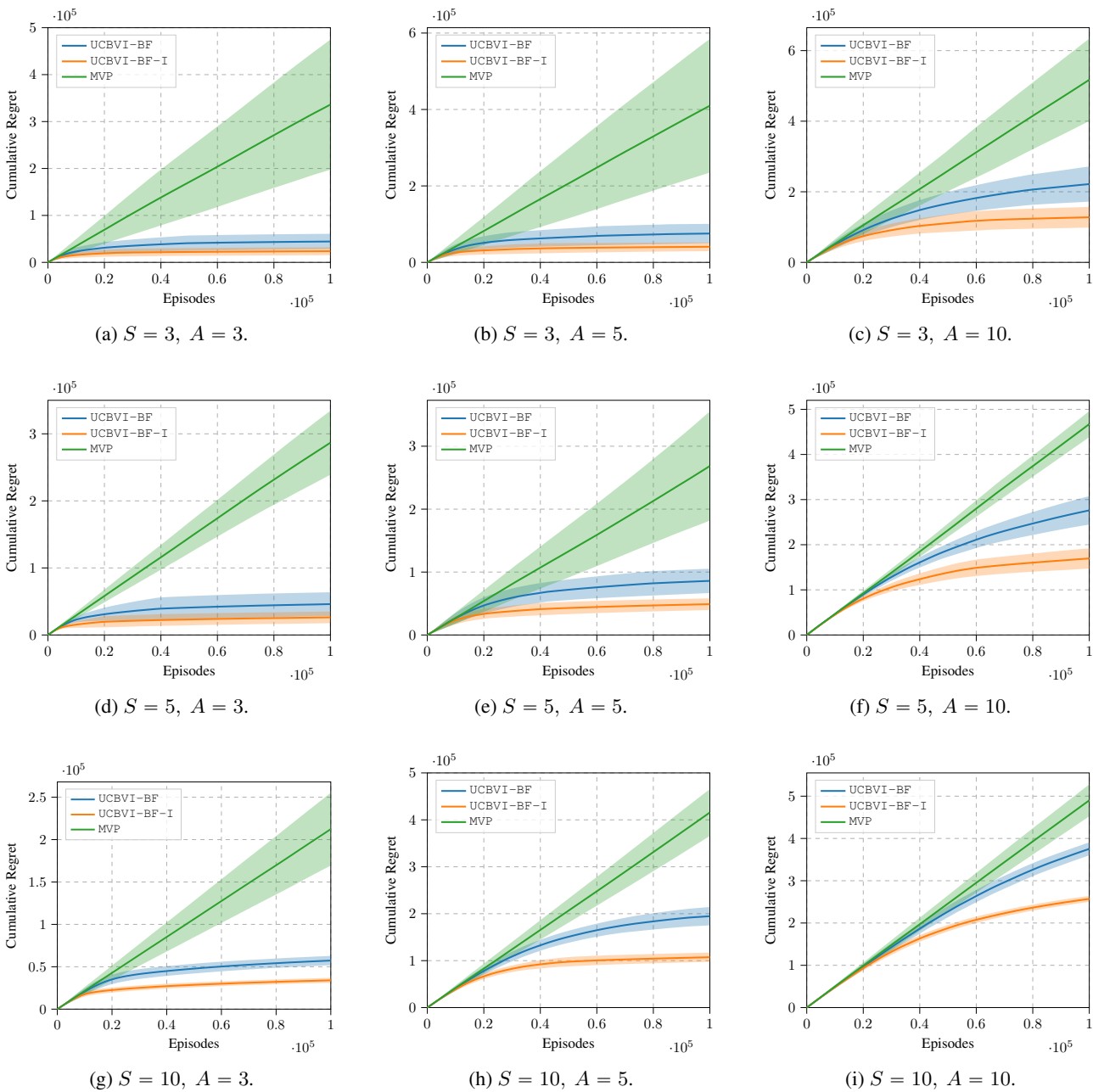

*Figure 5.* Performances in terms of cumulative regret in toy environments with $S \in \{3, 5, 10\}$ states and $A \in \{3, 5, 10\}$ actions for $H = 10$ and $K = 10^5$ (5 runs, mean $\pm$ 95% C.I.).

