# OpenReview forum: "Position: Constants are Critical in Regret Bounds for Reinforcement Learning"
_ICML.cc/2025/Position_Paper_Track — ICML 2025 Position Paper Track poster_

### Official Review · Reviewer_1WE5 · 2025-03-12

**Significance:** 3
**Argument Clarity:** 4
**Rating:** 3
**Confidence:** 3

**Questions:**

There are some questions for the authors.

* The authors show the importance of the multiplicative constants via numerical evaluations. Though the numerical studies proves the advantages of the UCBVI-bf method, it is not straightforward to show that the advantages come from the smaller constants. The reviewer believes the cumulative regret is not a good enough signal.

* For researchers proving the lower bound, the reason we discard the constants is that we assume that it may take a large number of interactions before the algorithm converges, leading the importance of the convergence orders instead of the constants.
However, in the numerical studies, it seems that $T$ is not large enough when the proposed method starts outperform the alternatives. Thus, it is not convincing enough that the multiplicative constants are truly crucial.

 * In addition, the authors considers the regret bound in bandit problems in section 3. However, the title of the manuscript is for Reinforcement Learning. The authors have not deliver a better algorithm (with lower constants) under real RL problems (stage-dependent). Thus, it is also convincing enough to claim the multiplicative constants are important for RL problem.

**Discussion Potential:**

3

**Paper Summary:**

This manuscript argues that the multiplicative constants in approaching the lower regret bound in bandit problems are very critical in practice. The authors first delvier the refined UCBVI algorithm (including UCBVI-ch and UCBVI-bf), both of which show better performance (equal order but better constants). Additionally, the authors  performed numerical evaluation proving the superiority of the refined algorithms, further demonstrating the constants are critical for approaching the lower bounds.

**Position:**

Yes

**Position In Title:**

Yes

**Related Work:**

3

**Strengths And Weaknesses:**

The manuscript is well-written and mathematically solid. The structure is very clear and the claim is solid.  It is appreciated that the author proposed two refined algorithms with better constants. However, the reasoning process is somehow weak. The reviewer can only realized the importance of the multiplicative constants when seeing the numerical evaluations. Actually, in section 3, it's difficult for us to recognize the importance of the constants.

**Support:**

3

---

> ### Author Rebuttal · Authors · 2025-04-01
>
> We thank the Reviewer for the time spent reviewing our work. Below, our answers to the Reviewer's comments and questions.
>
> > The reviewer can only realized the importance of the multiplicative constants when seeing the numerical evaluations. Actually, in section 3, it's difficult for us to recognize the importance of the constants.
>
> We agree with the Reviewer on the importance of the constants being more apparent when looking at the experimental evaluation. Nevertheless, constants play an important role in theoretical results. The goal of the regret analysis is to assess the performance of the algorithm in the worst-case instance. If we believe that the regret is a meaningful index to assess the performance of an algorithms (e.g., "to sell the algorithm to a customer"), then, all its components should be considered important. Indeed, suppose we have two algorithms that share the same dependence on the relevant quantities (such as $T$) but the former has a multiplicative constant of $1$, while the second has a multiplicative constant of $100$, can we really say that they are equivalent (e.g., "suppose regret is money")? Note that, for multiplicative constants, the gap is apparent even in the large sample regime $T \rightarrow + \infty$.
>
> > The authors show the importance of the multiplicative constants via numerical evaluations. Though the numerical studies proves the advantages of the UCBVI-bf method, it is not straightforward to show that the advantages come from the smaller constants. The reviewer believes the cumulative regret is not a good enough signal.
>
> The use of smaller constants in the bonus term of the UCBVI-BF-I algorithm modifies the trade-off between exploration and exploitation by reducing the unnecessary exploration. This can be observed at Line 14 of Algorithm 1, where the agent plays the action that maximizes the state-action value function. The estimation of the next state value function being equal, it is then clear that the selection of the agent depends on the bonus term. Indeed, since we have derived the bonus for the UCBI-BF-I algorithm following the same derivation of UCBVI-BF, the difference lies in the constants in the bonus term. As such, the improvement in performance shown is in the experiments is entirely ascribable to the better constant terms in the upper confidence bound.
>
> Regarding the use of the regret as a performance signal, we have selected it as it is the standard performance measure for online learning algorithms (Lattimore and Szepesvári, 2020).
>
> Lattimore, T. and Szepesvári, C. Bandit algorithms. Cambridge University Press, 2020.
>
> > For researchers proving the lower bound, the reason we discard the constants is that we assume that it may take a large number of interactions before the algorithm converges, leading the importance of the convergence orders instead of the constants. However, in the numerical studies, it seems that $T$ is not large enough when the proposed method starts outperform the alternatives. Thus, it is not convincing enough that the multiplicative constants are truly crucial.
>
> We remark that, when talking about constants, we refer to **multiplicative constants** which multiply the leading term of the regret. Thus, even in the limit of infinite interactions ($T \rightarrow \infty$), their effect is important on the final regret. This is confirmed by our experimental evaluation. We will clarify this in the paper.
>
> > In addition, the authors considers the regret bound in bandit problems in section 3. However, the title of the manuscript is for Reinforcement Learning. The authors have not deliver a better algorithm (with lower constants) under real RL problems (stage-dependent). Thus, it is also convincing enough to claim the multiplicative constants are important for RL problem.
>
> The UCBVI algorithm, revisited in this paper to derive tighter constants, is the first algorithm to match the lower bound for **tabular RL**, up to logarithmic terms. Let us restate the definition of the bonus term, for readability reasons:
>
> $$
> \\begin{align}
> b_{k,h}(x,a) = & \\underbrace{\sqrt{\frac{4L \\mathbb{V}\\mathrm{ar}\_{y \sim \widehat{P}\_k(\cdot | x,a)}(V_{k,h+1}(y))}{\max \\{ N_{k}(x,a), 1 \\}}}}\_{\\texttt{(A)}} + \\underbrace{\frac{7HL}{ 3 \max \\{ N_{k}(x,a) - 1, 1 \\}}}\_{\texttt{(B)}} \\\\
> & \quad + \underbrace{\sqrt{\frac{4 \sum_{y \in \mathcal{S}} \left( \widehat{P}\_k(y | x, a) \cdot \min \left\\{ \frac{84^2 H^3 S^2 A L^2}{\max \\{1, N_{k,h+1}'(y) \\} } , H^2 \right\\} \right)}{\max \\{ N_{k}(x,a), 1 \\}}}}_{\texttt{(C)}}.
> \\end{align}
> $$
>
> Our contribution regarding the constants in the bonus term is the following:
>
> - in term $\texttt{(A)}$, we have a $\sqrt{4}$ multiplicative factor instead of $\sqrt{8}$;
> - in term $\texttt{(B)}$, we have a $7$ multiplicative factor instead of $14$;
> - in term $\texttt{(C)}$, we have a $\sqrt{4}$ multiplicative factor instead of $\sqrt{8}$ and, inside the minimum, a term multiplicative factor $84^2$ instead of $100^2$.

---

> > ### Comment · Reviewer_1WE5 · 2025-04-06
> >
> > Thank you so much for your reply. I have updated some scores in my official review.

---

### Official Review · Reviewer_ni5b · 2025-03-13

**Significance:** 3
**Argument Clarity:** 4
**Rating:** 4
**Confidence:** 5

**Questions:**

Could you provide more examples where the loose constants in the regret bounds are detrimental to the empirical performance?

**Discussion Potential:**

3

**Paper Summary:**

The paper argues the importance of constants in regret bounds for online reinforcement learning algorithms. A tighter analysis of UCBVI leads to smaller constants in the bound and allows for less exploration, which in turn improves the empirical performance of the UCBVI variant (based on Chernoff-Hoeffding and Bernstein-Freedman inequalities). Additionally, the paper highlights that while MVP has favorable asymptotic bound, the doubling trick used leads to sample inefficiency and poor empirical performance.

**Position:**

Yes

**Position In Title:**

Yes

**Related Work:**

3

**Strengths And Weaknesses:**

Highlighting the role of the constants in the regret bounds is important, and focusing mostly on asymptotic properties can have detrimental effect on empirical (finite time) performance.

The theoretical derivations are valuable, although the case, probably, could have been made with results already available.

Overviewing a larger number of bounds where constants play an important role could have been useful.

**Support:**

4

---

> ### Author Rebuttal · Authors · 2025-04-01
>
> We thank the Reviewer for the time spent reviewing our paper. Below, our answers to the Reviewer's comments and questions.
>
>  > The theoretical derivations are valuable, although the case, probably, could have been made with results already available.
>
> As the Reviewer correctly observed, there are existing results which we could have used to support our arguments. However, our intention was to additionally remark that, given a state-of-the-art algorithm (i.e., UCBVI in 2017), one could explore two possible directions: (i) focus entirely on the theoretical aspect of the problem, in this case, removing the minimum $T$ for which the upper bound on the regret holds (i.e., MVP), or (ii) try to further improve the empirical performance of the algorithm, while preserving (some of) the theoretical guarantees.
>
> > Overviewing a larger number of bounds where constants play an important role could have been useful.
>
> We acknowledge that there exist works in the RL/bandit literature that go in the direction we advocate for, and we recognize the importance of reporting them. Examples of that are:
> - (Garivier and Cappé, 2011): improves bandit solutions for Bernoulli rewards, as also highlighted by Reviewer QdHu.
> - (Bubeck, 2010, Thr. 2.2): improves the analysis of (Auer et al., 2002) for the UCB1 algorithm, providing a more efficient bonus term, whose increment in performance is very evident in practice, even if the order of the regret remains the same (for details, we refer the Reviewer to the answer 3 we provide to Reviewer ni5b due to space constraints).
> - (Abbasi-Yadkori et al., 2011) improves the work of (Dani et al., 2008) on linear bandits, even if such improvement does not change the rate, but significantly improving the solution.
>
> We will revise the manuscript to discuss works like these.
>
> > Could you provide more examples where the loose constants in the regret bounds are detrimental to the empirical performance?
>
> An illustrative example of this is the difference between the two versions of the UCB1 algorithm. The original version (Auer et al., 2002) uses an argument based on Boole's inequality to derive the result. Later, (Bubeck, 2010, Thr. 2.2) developed a new version of UCB1 with a more convoluted analysis which allows to improve the confidence bound by a factor $4$ thanks to arguments based on Martingales and the Peeling argument.
> In the first case, the upper confidence bound contains a $\sqrt{2 \log(T) / n}$ term, where $T$ is the time horizon and $n$ is the number of times an arm has been pulled. In the latter, such bound contains a $\sqrt{\alpha \log (t) / n}$, where $t$ is the current time step, and the regret bound is provided for any $\alpha > 1/2$. This modification reduces the unnecessary exploration, thus improving the performance of the algorithm.
> Nevertheless, this improvement cannot be observed in the order of the regret of the two versions of the algorithm, which is equal from both the minimax and instance-dependent perspective (up to constants), but this refinement in the upper confidence bound allows to significantly increase the empirical performance of the algorithm.
> In support of this argument, we designed an experiment to evidence it when we vary the number of actions $A \in \\{ 3,5,10 \\}$ and the time horizon $T\in\\{ 10^4, 5 \cdot 10^4, 10^5 \\}$. We refer the Reviewer to the following link https://drive.google.com/file/d/1URA2T-9iv3h7peUSs-_718lj5o4GL1fu/view?usp=sharing (anonymized) for the plots of the experiments.
> We will add this discussion in the paper on this example.
>
> ---
>
> Garivier, A. and Cappé, O., The KL-UCB Algorithm for Bounded Stochastic Bandits and Beyond. Annual Conference on Learning Theory, 2011.
>
> Bubeck, S. Bandits games and clustering foundations. PhD diss., Université des Sciences et Technologie de Lille-Lille I, 2010.
>
> Auer, P., Cesa-Bianchi, N., and Fischer, P. Finite-time analysis of the multiarmed bandit problem. Machine Learning, 2002.
>
> Abbasi-Yadkori, Y., Pál, D., and Szepesvári, C. Improved algorithms for linear stochastic bandits. Advances in Neural Information Processing Systems, 2011.
>
> Dani, V., Hayes, T. P., and Kakade, S. M. Stochastic linear optimization under bandit feedback. Annual Conference on Learning Theory, 2008.

---

> > ### Comment · Reviewer_ni5b · 2025-04-03
> >
> > I think the suggestions in the rebuttal will provide a broader view on the importance of the constants.

---

### Official Review · Reviewer_AHpx · 2025-03-15

**Significance:** 3
**Argument Clarity:** 4
**Rating:** 5
**Confidence:** 3

**Questions:**

- Would there be any potential downside in considering lower-order terms and constants when designing practical algorithms? Could the authors offer any insights on potential tradeoffs (*e.g.*, computational overhead)?
- Can similar refinements with constants be applied beyond the finite-horizon tabular RL (*e.g.*, to settings with function approximation or model-free RL)? Could the authors suggest any concrete future directions?

**Discussion Potential:**

3

**Paper Summary:**

This paper advocates *not* to ignore the constants in regret bounds for reinforcement learning.

Specifically, the paper uses the Bernstein-Freedman inequality to derive a smaller exploration bonus (in terms of its constant coefficient), which (i) reduces unnecessary exploration, (ii) results in a tighter regret bound with a smaller constant factor (while the asymptotic order is the same), and (iii) leads to a significant empirical improvement in two synthetic MDP environments.

**Position:**

Yes

**Position In Title:**

Yes

**Related Work:**

3

**Strengths And Weaknesses:**

Strengths:
- The paper is very well-written and easy-to-follow.
- The perspective is fresh and can potentially be very impactful. While much of the field focuses on asymptotic order, the authors argues for the perspective that constants can have a substantial impact on practical performance. I believe the arguments and evidence presented in this paper would serve as good references to future researchers.
- The position is well supported by both theoretical and empirical evidence.


Weaknesses:
Overall, I think this is a good position paper. There are some parts that authors may further improve, for example, the current experimental validation may be a bit limited. It would be nice to see the results can remain valid with larger environments, or on a broader set of RL algorithms.

**Support:**

3

---

> ### Author Rebuttal · Authors · 2025-04-01
>
> We thank the Reviewer for the time spent reviewing our paper and for appreciating the position we advocate for. Below, our answers to the Reviewer's comments.
>
>  > There are some parts that authors may further improve, for example, the current experimental validation may be a bit limited. It would be nice to see the results can remain valid with larger environments, or on a broader set of RL algorithms.
>
> Regarding the effect of tighter constants with varying environment sizes, we refer the Reviewer to the additional experiments at the following link https://drive.google.com/file/d/16yorH6BpPSTOZNNkhj4BD48Pj2HFrFOC/view?usp=sharing, provided via an anonymized account.
> At the link, the Reviewer can find plots of experiments comparing the UCBVI-BF, UCBVI-BF-I, and MVP algorithms, in environments with state space and action space with cardinalities $S, A \in \\{3, 5, 10\\}$, time horizon $H = 5$ (Figure 3) and $H = 10$ (Figure 4), and $K= 10^5$ episodes, averaged over 5 runs, with a $95\\%$ confidence interval. As can be seen in the experiments, the performance improvement of UCBVI-BF-I remains evident even as the complexity of the environment increases. We added and commented on these additional experiments in the paper.
>
> > Would there be any potential downside in considering lower-order terms and constants when designing practical algorithms? Could the authors offer any insights on potential tradeoffs (e.g., computational overhead)?
>
> Designing algorithms with the goal of controlling (multiplicative) constants and lower-order terms may lead to: (i) less intuitive algorithmic solutions, with possible larger computational burden and (ii) more intricated analysis. The case of multi-armed bandits is emblematic. Indeed, the most intuitive (and computationally very efficient) UCB1 algorithm, that enjoys a very clean regret analysis, does not match neither the instance-dependent lower bound, since it has a larger multiplicative constant, nor the minimax lower bound, because of an additional $\sqrt{\log T}$ term. To fix the first issue, in Chapter 8 of Lattimore and Szepesvári, 2020, Asymptotically Optimal UCB  is proposed which uses a more complex exploration index. Similarly, to fix the second issue, Chapter 9 of Lattimore and Szepesvári, 2020 presents the MOSS algorithm that, similarly, leverages a different exploration index. Although the computational complexity does not increase significantly, in both cases, the resulting analysis is more intricate and less clean compared to UCB1.
> We will insert a comment on this matter in the paper.
>
> Lattimore, T. and Szepesvári, C. Bandit algorithms. Cambridge University Press, 2020.
>
> > Can similar refinements with constants be applied beyond the finite-horizon tabular RL (e.g., to settings with function approximation or model-free RL)? Could the authors suggest any concrete future directions?
>
> We conjecture that it could indeed be possible to apply such refinements beyond the setting we considered in our discussion. One important remark, however, is that such settings (e.g., the ones suggested by the Reviewer) are harder to analyze, and the scientific community may not yet possess all the necessary tools to perform the analyses.
> Two concrete future directions, in our opinion, are (i) to try to study the theoretical guarantees of existing algorithms, resulting in a deeper knowledge of the settings and in turn allowing the definition of better performing algorithms, and (ii) to research novel mathematical tools, which may enable the analysis of complex settings. We will add a comment on this in the paper.

---

### Official Review · Reviewer_QdHu · 2025-03-20

**Significance:** 2
**Argument Clarity:** 3
**Rating:** 4
**Confidence:** 4

**Questions:**

This paper makes a valuable theoretical contribution by highlighting the importance of constants in regret minimization. However, its message is not as original as the paper claims, and is somewhat aggressively expressed at some points.
Moreover, the authors did not seriously list all the articles sharing the same care for constants.
In a nutshell, this is a thought-provoking read for RL theorists and practitioners that would need some corrections and complements before publication.


The writing quality of this paper is clear and well-structured but has some readability issues due to its dense mathematical content and occasional awkward phrasing.
In particular, the proofs in appendix are lengthy and it would be usefule to help the reader by providing a proof sketch or at least a structure in the proof


    "Ignoring multiplicative constants when evaluating whether an algorithm matches the problem’s lower bound may lead to disappointing results when using such an algorithm."
    "when using such an algorithm" is redundant and wordy.
    Suggestion: "Ignoring multiplicative constants when evaluating whether an algorithm matches the problem’s lower bound may lead to disappointing practical performance."


    "Starting from the well-known UCBVI algorithm, we improve  UPON the bonus terms and the corresponding regret analysis."


    "Constants matter in two key moments: (i) Algorithm design: the exploration strategy of the algorithm should be designed to enforce the minimal required exploration needed in order to achieve the desired regret performance."
    Issue: "the minimal required exploration needed" is redundant.



    "This reduction in over-exploration has significant empirical effects, as shown in the experiments in Section 4, where, as reported in Table 1, we achieve an improvement in the empirical regret of 1.87 times."
-> "This reduction in over-exploration has significant empirical effects, as shown in the experiments in Section 4, where we achieve a 1.87× reduction in empirical regret."

 Lemma C.1 : very classical better bound, see e.g. https://www.stat.cmu.edu/~larry/=sml/Concentration.pdf Lemma 7.37

**Discussion Potential:**

3

**Ethical Review Concerns:**

The concerns on some results in cited papers are expressed in a somewhat rude manner

**Paper Summary:**

This paper challenges the alleged mainstream theoretical research in reinforcement learning (RL), which focuses on regret bounds that match lower bounds up to multiplicative constants. The authors argue that these constants play a significant role in practical performance and should not be ignored. Using finite-horizon tabular Markov Decision Processes (MDPs) as a case study, they refine the UCBVI algorithm by improving the bonus terms and regret analysis. Empirical results show that reducing multiplicative constants leads to a significant improvement in empirical regret (1.87× reduction on average). The paper calls for a better alignment between theoretical guarantees and practical performance in RL research.

Key Claims and Their Supporting Evidence
    Ignoring multiplicative constants leads to poor empirical performance.
    A refined version of UCBVI can improve regret bounds without changing the order.
    Current theoretical RL research is misaligned with practice.

 Methods and Evaluation Criteria
        The authors use cumulative regret as the primary evaluation metric.
        They compare theoretical regret bounds with empirical results.
        The paper introduces tighter exploration bonuses and analyzes their impact on regret minimization.
        The authors compare their refined UCBVI with:
            The original UCBVI (Azar et al., 2017)
            The MVP algorithm (Zhang et al., 2024)
        The effectiveness of reducing constants is tested using synthetic environments (e.g., RiverSwim).

Theoretical Claims
        The authors optimize bonus terms in UCBVI using Bernstein-Freedman and Chernoff-Hoeffding inequalities.
        This results in lower exploration costs and tighter regret bounds.
        The new regret bound is of the same order as previous ones but has smaller constants, making it empirically superior.
        The authors show that MVP discards samples inefficiently, leading to worse empirical performance despite theoretical guarantees.

Content of the Supplementary Material
    Detailed proofs of regret bounds (refining Azar et al., 2017).
    Algorithm pseudocode (UCBVI with refined bonuses).
    Mathematical derivations of concentration inequalities used in the analysis.
    Extended discussions on previous regret analyses.


## update after rebuttal
I have read the discussion and suggest acceptance.

**Position:**

Yes

**Position In Title:**

Yes

**Related Work:**

2

**Strengths And Weaknesses:**

Strengths

-I totally share the message: constants are important, matching a rate is not a full answer
- the improved analysis is interesting
- the numerical experiments are relevant to illustrate the issue

Weaknesses:
- No real-world applications: All experiments are on synthetic benchmarks; the impact on deep RL or real-world decision-making is unclear.
- the paper does not cite all the previous works that go in the same direction. The message of the paper (the importance of constants) was for example well illustrated by [The KL-UCB Algorithm for Bounded Stochastic Bandits and Beyond Aurélien Garivier, Olivier Cappé. Proceedings of the 24th Annual Conference on Learning Theory, PMLR 19:359-376, 2011. ] .

**Support:**

3

---

> ### Author Rebuttal · Authors · 2025-04-01
>
> We would like to thank the Reviewer for the time spent reviewing our work. Below, we address the Reviewer's concerns and questions.
>
> > No real-world applications: All experiments are on synthetic benchmarks; the impact on deep RL or real-world decision-making is unclear.
>
> We have selected a setting (i.e., finite-time, tabular RL) which we deemed illustrative of the problem we wished to discuss in the paper and we provided two examples, although synthetic, that allow for an evaluation of algorithms both in a simple-but-relevant and in a challenging environment (i.e., RiverSwim). If the difference in performance ascribable to tighter constants is considerable in tabular RL environments, we conjecture that it will be exacerbated when considering the much more complex setting of Deep RL. Additionally, tabular RL is well suited for this kind of discussion, as it is easier to define algorithms with theoretical guarantees than in Deep RL, where the necessary analyses are more complex. We will add a remark on this point in the paper. Indeed, when moving to algorithms for Deep RL endowed with theoretical guarantees, it is often the case that the anaysis becomes more intricate and it is more prone to suboptimalities where the attention to the constants is less considered.
>
> > the paper does not cite all the previous works that go in the same direction. The message of the paper (the importance of constants) was for example well illustrated by (Garivier and Cappé, 2011).
>
> We acknowledge that there exist works in the RL literature that go in the direction we advocate for, and we recognize the importance of reporting them. As the Reviewer correctly reported, (Garivier and Cappé, 2011) is an important example in this direction. Additional examples in our opinion are:
> - (Bubeck, 2010, Thr. 2.2): improves the analysis of (Auer et al., 2002) for the UCB1 algorithm, providing a more efficient bonus term, whose increment in performance is very evident in practice, even if the order of the regret remains the same (for details, we refer the Reviewer to the answer 3 we provide to Reviewer ni5b due to space constraints).
> - (Abbasi-Yadkori et al., 2011) improves the work of (Dani et al., 2008) on linear bandits, even if such an improvement does not change the rate, but significantly improves the solution.
>
> We will revise the manuscript to discuss these works in the main paper as *virtuous* examples of the importance of deriving tighter constants and lower order terms. We thank the Reviewer for this important observation.
>
> Bubeck, S. "Bandits games and clustering foundations." PhD diss., Université des Sciences et Technologie de Lille-Lille I, 2010.
>
> Auer, P., Cesa-Bianchi, N., and Fischer, P. "Finite-time analysis of the multiarmed bandit problem." Machine Learning, 2002.
>
> Abbasi-Yadkori, Y., Pál, D., and Szepesvári, C. Improved algorithms for linear stochastic bandits. Advances in Neural Information Processing Systems, 2011.
>
> Dani, V., Hayes, T. P., and Kakade, S. M. Stochastic linear optimization under bandit feedback. Annual Conference on Learning Theory, 2008.
>
>
> > This paper makes a valuable theoretical contribution by highlighting the importance of constants in regret minimization. However, its message is not as original as the paper claims, and is somewhat aggressively expressed at some points. Moreover, the authors did not seriously list all the articles sharing the same care for constants. In a nutshell, this is a thought-provoking read for RL theorists and practitioners that would need some corrections and complements before publication.
>
> In our opinion, the paper does not employ aggressive language when commenting on existing algorithms and on our numerical validations, but rather tries to highlight those aspects in which theory-driven and application-driven approaches are most distant. We kindly ask the Reviewer for detailed indications of which parts they found aggressively written, to allow us to thoroughly review them.
>
> > The writing quality of this paper is clear and well-structured but has some readability issues due to its dense mathematical content and occasional awkward phrasing. In particular, the proofs in appendix are lengthy and it would be usefule to help the reader by providing a proof sketch or at least a structure in the proof
>
> Thank you for the all the suggestions aimed at improving the readability of our work. We commit to  modify the structure of the paper, in order to include the additional examples cited above in the main paper, thus improving the presentation and readability, and to remove any unnecessary mathematical content from the main paper, moving it to the appendix together with a structure of the proofs provided.
> Also, thank you for the reference regarding Lemma C.1.

---

### Decision · Program_Chairs · 2025-04-30

**Decision:**

Accept (poster)

**Comment:**

All reviewers concurred that the paper's central argument—that reducing the comparison of Reinforcement Learning (RL) methods to mere regret rates can be misleading—is relevant and likely to stimulate constructive debates within the community. The reviewers appreciated the use of a specific, worked out example to support this argument. However, some reviewers noted that this approach may limit the paper's generality. They suggested that the paper could benefit from incorporating elements from broader discussions found in previously published works and included several other suggested improvements related to the wording and clarity of the paper.